# Parallel interrogation of the chalcogenide-based micro-ring sensor array for photoacoustic tomography

Jingshun Pan[1,2,3,4], Qiang Li[1,4], Yaoming Feng[1], Ruifeng Zhong[1], Zhihao Fu[1], Shuixian Yang[1], Weiyuan Sun[1], Bin Zhang[1,2], Qi Sui[2], Jun Chen[1], Yuecheng Shen[1,2] ✉ & Zhaohui Li[1,2] ✉

Photoacoustic tomography (PAT), also known as optoacoustic tomography, is an attractive imaging modality that provides optical contrast with acoustic resolutions. Recent progress in the applications of PAT largely relies on the development and employment of ultrasound sensor arrays with many elements. Although on-chip optical ultrasound sensors have been demonstrated with high sensitivity, large bandwidth, and small size, PAT with on-chip optical ultrasound sensor arrays is rarely reported. In this work, we demonstrate PAT with a chalcogenide-based micro-ring sensor array containing 15 elements, while each element supports a bandwidth of 175 MHz (−6 dB) and a noise-equivalent pressure of 2.2 mPaHz$^{-1/2}$. Moreover, by synthesizing a digital optical frequency comb (DOFC), we further develop an effective means of parallel interrogation to this sensor array. As a proof of concept, parallel interrogation with only one light source and one photoreceiver is demonstrated for PAT with this sensor array, providing images of fast-moving objects, leaf veins, and live zebrafish. The superior performance of the chalcogenide-based micro-ring sensor array and the effectiveness of the DOFC-enabled parallel interrogation offer great prospects for advancing applications in PAT.

Biomedical imaging serves as an effective tool in life science to explore the mystery of behaviors and recognitions of humans and animals. Among various biomedical imaging modalities, photoacoustic (PA) tomography (PAT), also known as optoacoustic tomography, is a hybrid imaging modality that combines optical excitation and ultrasonic detection[1,2]. This marriage overcomes the disadvantages of low contrast in ultrasonic imaging and shallow depth in optical imaging, making PAT a promising tool for deep-tissue high-resolution structural and functional imaging in vivo. Currently, PAT has been demonstrated with great success with many preclinical and clinical applications, such as whole-body imaging for small animals[3], functional imaging for mouse and human brains[4,5], and human breast imaging for cancer screening[6,7]. In these demonstrations, sensor arrays that contain many commercial piezo-electric ultrasound sensors were generally adopted. However, these bulk sensors suffer from limited acceptance angle, sensitivity, and frequency bandwidth, degrading imaging quality and causing reconstruction artifacts. Moreover, high-framerate three-dimensional PAT requires a large number of elements, leading to huge system complexity with complicated electrical interconnects. To overcome this challenge, optical ultrasound sensors with a small footprint, large acceptance angle, high sensitivity, and broad frequency bandwidth are becoming promising substitutes[8].

A variety of optical ultrasound sensors have been proposed to detect pressure-induced optical changes, which reflect the amplitude of

[1]School of Electronics and Information Technology, Guangdong Provincial Key Laboratory of Optoelectronic Information Processing Chips and Systems, Sun Yat-sen University, Guangzhou 510275, China. [2]Southern Marine Science and Engineering Guangdong Laboratory (Zhuhai), Zhuhai 519000, China. [3]Guangdong Provincial Key Laboratory of Nanophotonic Functional Materials and Devices, South China Normal University, Guangzhou 510006, China. [4]These authors contributed equally: Jingshun Pan, Qiang Li. ✉e-mail: shenyuecheng@mail.sysu.edu.cn; lzhh88@mail.sysu.edu.cn

the acoustic waves[8]. Thanks to the abundant means to address optical signals, optical ultrasound sensors have now shown superior performances in different aspects with interferometric methods. For example, fiber-optic interferometers can serve as high-performance ultrasound sensors, bringing excellent flexibility to PAT by enabling different imaging configurations[9–11]. Moreover, fiber tips with engineered micro-resonators, such as in-fiber Bragg grating[12] and Fabry-Perot structure[13–15], can also be made as sensitive ultrasound probes to image vascular structure in endoscopy[16]. Moreover, by exploiting the platform of silicon photonics, miniaturized ultrasound sensors based on Bragg grating can be made with a small footprint[17–19], even down to the size of a sub-micrometer[20,21]. Alternatively, micro-ring resonators have the advantages of high-quality factor, miniaturized form factor, and optical transparency[22–26]. To date, optical ultrasound sensors that employ either polymer-based[22,23] or silicon-based[27] micro-rings can achieve a high-quality factor of up to $10^5$ with a small footprint[28,29], an ultrabroad frequency bandwidth of up to 350 MHz[30], high sensitivity with a noise-equivalent pressure (NEP) down to 1.3 mPaHz$^{-1/2}$[27], and a wide acceptance angle over ±30 degrees[25]. Using micro-ring ultrasound sensors, PAT with laser-scanning endoscopy on tube phantoms[31] and microscopy on single cells[32], mouse ears[29], and mouse brains[33] have been demonstrated. Recently, three-dimensional PAT for a variety of biological tissue with an imaging depth of up to 8 mm has also been reported[34].

Despite these accomplishments, these demonstrations on PAT mainly employ only a single element, which means tomographic images require a time-consuming scanning process. In contrast, PAT equipped with an ultrasound sensor array with many elements can facilitate imaging speed and improve imaging quality. Etalon sensors based on Fabry-Perot cavities are naturally two-dimensional sensor arrays[35,36]. However, parallel interrogation requires photodetector arrays with increased electrical complexity and suffers from considerable crosstalk. Similarly, fiber-optic detector arrays also lead to a rather complicated imaging system[10,37,38]. Back in 2008, Maxwell et al. reported the first integrated on-chip sensor array with four micro-rings[22]. The relatively low-quality factor of the micro-rings fabricated at that time causes spectrum overlapping between adjacent resonant

frequencies, making each element in the array difficult to function simultaneously. Recently, a one-dimensional sensor array containing ten micro-ring sensors was reported with a well-separated spectrum for adjacent resonant frequencies[27]. However, as this system had only one laser and one photoreceiver, only one micro-ring sensor can work at a time. It is expected that parallel interrogation could be realized by the means of wavelength division multiplexing[37], but it either requires a sufficient number of source-detector pairs[27] or a frequency-swept source[38]. Simultaneous multi-channel ultrasonic detection of up to four elements was reported via phase-modulated pulse[39], but it still requires one photoreceiver per detecting channel, which increases system complexity. Thus, there is an urgent need to not only fabricate compact and integrated on-chip optical ultrasound sensor arrays with a large number of elements but also develop efficient means to interrogate them in parallel.

To fill this blank, we reported a micro-ring sensor array in this work, which contains 15 high-quality micro-rings coupled to a bus straight waveguide. These structures were made of chalcogenide materials with large elastic-optical coefficients and low Young's modulus. Specifically, these micro-rings were characterized with high-quality factors ranging from $5 \times 10^5$ to $7 \times 10^5$, corresponding to ultrasonic sensors with a frequency bandwidth up to 175 MHz (−6 dB), a NEP down to 2.2 mPaHz$^{-1/2}$ or 7.1 Pa within a 20-MHz range, an acceptance angle of ±30 degrees (−3 dB at 25 MHz), and a small footprint of $0.85 \times 40 \times 40 \ \mu m^3$. To realize simultaneous operation, these micro-rings were delicately tuned with slightly different resonant frequencies. Assisted by a delicately synthesized digital optical frequency comb (DOFC), we further developed an effective means of parallel interrogation to the sensor array, which only requires one laser source and one photoreceiver in the system to perform PAT. In contrast to optically generated frequency combs, the DOFC holds the unique advantage of generating an ultra-narrow and tunable comb tooth. Since on-chip micro-rings with high-quality factors generally exhibit resonant dips with narrow linewidth, this unique property of the DOFC is well suited to locate resonant frequencies for all micro-ring sensors in parallel with high accuracy. As schematically shown in Fig. 1, the

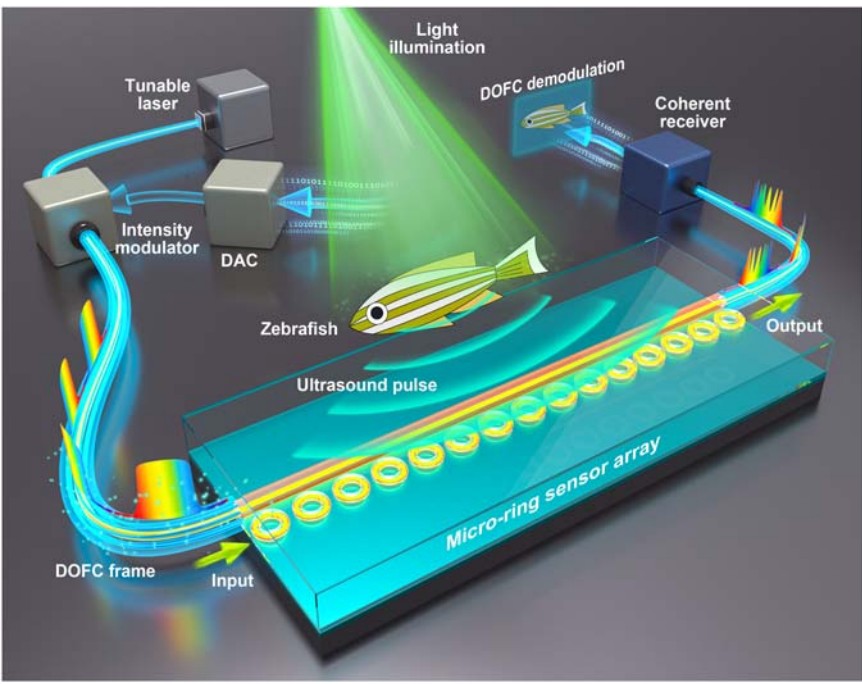

**Fig. 1 | A schematic illustration of performing photoacoustic tomography (PAT) with the chalcogenide-based micro-ring sensor array.** A digital optical frequency comb (DOFC) enabled parallel interrogation allows acoustic signals at all micro-ring sensors to be acquired using only one light source and one photoreceiver in a one-time measurement. DAC digital-to-analog converter.

DOFC passes through the sensor array and acquires the information of the laser-induced PA signals, enabling one-time measurement for the transmission spectrum using only a single photoreceiver. The knowledge of the complete transmission spectrum allows the determination of the PA signals received by all micro-rings simultaneously, allowing the formation of the images in a single-shot measurement. Equipped with the chalcogenide-based micro-ring sensor array and the means of parallel interrogation, we experimentally demonstrated PAT on fast-moving objects like a laser-scanning hot spot and a water-pulled microsphere. We further performed PAT with broad beam illumination on leaf veins buried inside thick tissue-mimicking phantom and live zebrafish at various growth stages. These results indicate that the marriage of these two technical innovations, i.e., high-performance micro-ring sensor array and DOFC-enabled parallel interrogation means, breeds a compact imaging system of PAT using an ultrasound optical sensor array, offering great prospects for clinical applications.

## Results

### Experimental results on imaging fast-moving objects in a single-shot measurement

We first implemented parallel interrogation to the sensor array on imaging fast-moving objects. In the first demonstration, a fast-moving hot spot was created by scanning a focused light pulse on a black tape through a two-axis galvo mirror. The focused pulse light was generated by the same laser used for sensor characterization, which has a repetition rate of 10 kHz. The hot spot, which has a diameter of about 100 μm and an energy of 1 nJ, moved along a trajectory of an '8' shape. The DOFC enables parallel interrogation of the PA signals for all micro-ring sensors simultaneously, and universal back-projection (UBP)[40] was employed to reconstruct the image for this linear array. Although the frame rate under this imaging scheme can be up to 10 kHz, we pick only eight representative images within a time window of 5 ms to reveal the moving trajectory of the created hot spot in Fig. 2(a). An outline of '8' is provided in white for visualization purposes.

Besides imaging a virtual hot spot, we also demonstrated imaging a physically presented polystyrene microsphere in motion. The microsphere has a diameter of 200 μm and was placed inside a plastic tube with a 500-μm inner diameter. The plastic tube was pumped with

flowing water, pushing the microsphere to move downward. In this demonstration, a 532-nm Nd: YAG laser (Beamtech, Dawa 100) with a pulse width of 6.5 ns and a repetition rate of 20 Hz was employed to illuminate the region of interest. A ground glass diffuser (Thorlabs, DG10-120) expanded the pulsed light (1.4 mJ) with a fluence rate of about 19.7 mJ/cm² and an illumination spot with a 3-mm diameter. Images reconstructed at different time intervals are shown in Fig. 2(b), exhibiting good reliability in imaging moving objects using the micro-ring sensor array with parallel interrogation. By quantifying the traveling distance within a given time interval, the moving speed of the microsphere was estimated to be 4.8 mm/s, which is close to the speed of flowing water in the tube.

### Experimental result on imaging biological samples

We further applied the micro-ring sensor array with parallel interrogation on imaging biological samples with complex structures. To mitigate unwanted artifacts due to limited view, biological samples were mounted on a motorized rotational stage. The scanning step size was set to 1 degree during the experiment. No additional averaging was required for imaging these biological samples. This circular scanning geometry endows the sensor array with a full view of biological samples, which effectively reduces the occurrence of reconstruction artifacts. Both the biological sample and the micro-ring sensor array lie in the same horizontal plane, which is detailed and sketched in Supplementary Note 1. The first biological sample to be imaged is a piece of leaf buried inside a 5-mm-thick tissue-mimicking phantom (transport mean free path ~1 mm⁻¹ at 532 nm). The preparation process of the leaf can be found in Supplementary Note 2. The fluence rate on the surface of the phantom was about 7 mJ/cm² and the illumination area has a diameter of about 8 mm. Figure 3(a) shows the reconstructed image of the leaf using the micro-ring sensor array, showing clear vein structures along various directions. As a comparison, the reconstructed image using a typical micro-ring sensor is shown in Fig. 3(b). More reconstructed images using other single micro-ring sensors are provided and compared in Supplementary Note 3, exhibiting similar performance in such a rotation-based detection geometry. Notably, leaf veins in Fig. 3(a) exhibit a much better contrast than that in Fig. 3(b). In particular, enlarged images of the part enclosed in the

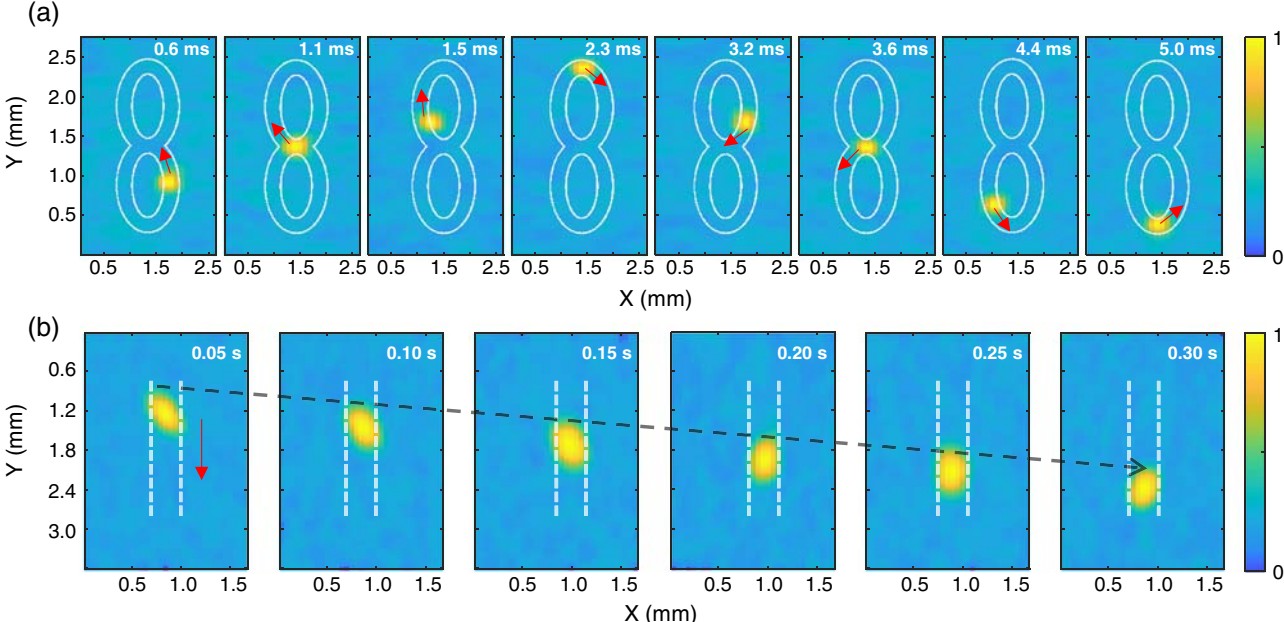

**Fig. 2 | Experimental results of imaging fast-moving objects using the micro-ring sensor array in a single-shot measurement with parallel interrogation.** **a** Representative images of a fast-moving hot spot along a trajectory of an '8' shape. **b** Imaging results for a polystyrene microsphere moving in motion.

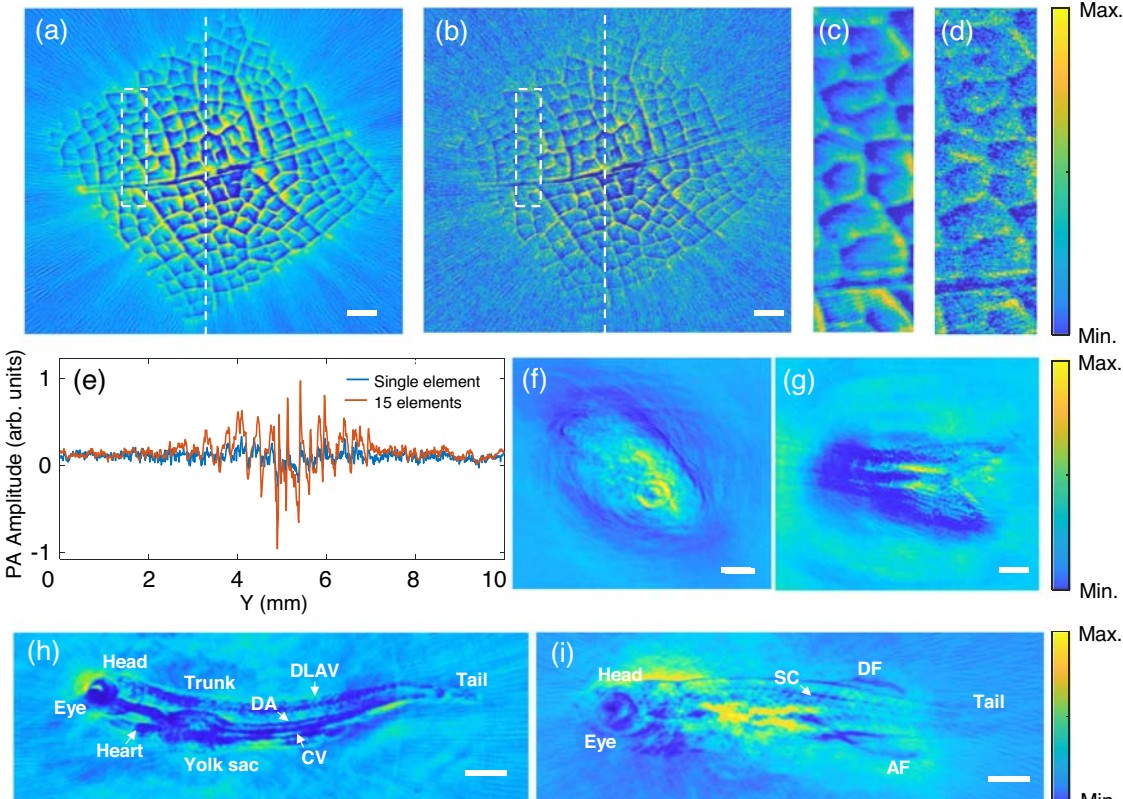

**Fig. 3 | Experimental results of imaging biological sample using the micro-ring sensor array with parallel interrogation.** Scale bars: 1 mm. **a** Imaging result of a piece of leaf buried inside 5-mm thick tissue-mimicking phantoms using the micro-ring sensor array. **b** imaging result of the same leaf using only one micro-ring sensor. **c, d** Enlarged view of the area enclosed in white dashed boxes in (**a** and **b**). **e** One-dimensional profiles along the white lines in the reconstructed images in (**a** and **b**) are plotted in red and blue, respectively. **f** Reconstructed image of the head of a 3-month-old adult zebrafish using the micro-ring sensor array. **g** Reconstructed image of the tail of a 3-month-old adult zebrafish using the micro-ring sensor array, (**h**) Whole-body imaging of a zebrafish 7 days post-fertilization using the micro-ring sensor array. DA dorsal aorta, CV cardinal vein, DLAV dorsal longitudinal anastomotic vessel. **i** Whole-body imaging of a zebrafish 20 days post-fertilization using the micro-ring sensor array. SC spinal column, DF dorsal fin, AF anal fin.

white boxes of Fig. 3(a) and (b) are also shown in Fig. 3(c) and (d), respectively. To quantify the contrast-to-noise ratio (CNR) of these images, Fig. 3(e) plots the signal strength along the white dashed lines in Fig. 3(a) and (b). Here, CNR is defined as the ratio of the peak value and the standard deviation of the background (away from the center). The improvement in CNR by using the micro-ring array compared to a single micro-ring is about 5.51 dB, which is close to the theoretical enhancement due to the element number ($\sqrt{15}$). These results confirm the superiority of the micro-ring sensor array over a single micro-ring sensor.

We also performed PAT for live zebrafish at the different growth stages, by using the DOFC-enabled parallel interrogation for the micro-ring sensor array. The zebrafish was first anesthetized using water with 0.016% tricaine and then moved to the imaging area. To minimize the movement of the live zebrafish, its abdomen was covered with 1% low-melting-point agar. All experiments using zebrafish were conducted under the auspices of animal ethics. The general breeding and handling of zebrafish were conducted following standard protocols from the Institutional Animal Care Committee of Sun Yat-sen University. As mentioned above, the fluence rate was about 7 mJ/cm², which is much lower than the American National Standards Institute (ANSI) safety limit. For a 3-month-old adult zebrafish, the reconstructed images for its head and tail are shown in Fig. 3(f) and (g), respectively. We also performed whole-body imaging for smaller zebrafish that were 7 days and 20 days post-fertilization, and the reconstructed images are shown in Fig. 3(h) and (i), respectively. As we can see from the figure, both the outlines of the zebrafish and their organs like the eye, heart, trunk, yolk sac, dorsal aorta, cardinal vein, dorsal longitudinal anastomotic vessel, spinal column, dorsal fin, and anal fin can be identified. These imaging results demonstrate the feasibility of the micro-ring sensor array, as well as the effectiveness of the DOFC-enabled parallel interrogation, for PAT in vivo. It is also noticeable that the imaging resolutions of relatively thick biological samples do not look as good as the ones characterized above. Such an observation is likely due to the strong attenuation of high-frequency ultrasound inside thick samples. This fact indicates that for deep-tissue imaging, one may sacrifice the bandwidth of the sensor to gain benefits in other properties.

Optical Sensors have the advantage of working in the planar geometry[41]. Here, instead of rotating samples, we show that our technique can function in the planar geometry as well. As shown in Fig. 4(a) (camera-captured image), three interleaved black hairs are chosen as the imaging target and are buried inside tissue-mimicking phantoms. Since 15 elements are not enough to suppress reconstruction artifacts in PAT, we linearly scanned the sensor array within a ± 8-mm range and with a 20-μm step size. The reconstructed image is shown in Fig. 4(b), showing excellent agreement with the original layout displayed in Fig. 4(a). Moreover, Fig. 4(c), (d), (e), and (f) show the reconstructed images using the information collected only within the range of (−8, −4) mm, (−4, 0) mm, (0, 4) mm, and (4, 8) mm, respectively. These results demonstrate that in contrast to the rotational geometry, different sensors in the planar geometry contain information through different projections.

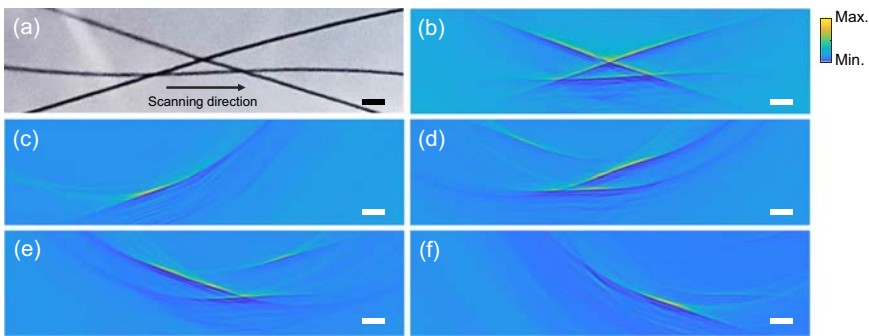

**Fig. 4 | Experimental results of imaging interleaved black hairs in the planar geometry using the micro-ring sensor array with parallel interrogation.** Scale bars: 1 mm. **a** Camera-captured image of the layout of three interleaved black hairs. **b** Reconstructed image by linearly scanning the sensor within a ± 8-mm range and with a 20-μm step size. **c**–**f** Reconstructed images using the information collected within the range of (−8, −4) mm, (−4, 0) mm, (0, 4) mm, and (4, 8) mm, respectively.

## Discussion

In this work, a micro-ring sensor array containing 15 elements was demonstrated for PAT with DOFC-enabled parallel interrogation. In our framework, the limiting factor on the allowable sensor number is the bandwidth of the DOFC rather than the fabrication technique. DOFC-enabled parallel interrogation requires the resonant peaks of all micro-rings to be identifiable. Given the 40-GHz bandwidth of the DOFC and the 1.66-GHz separation of the resonant frequencies, at most 24 micro-rings can be accommodated theoretically. Given practical concerns such as crosstalk and the edge effect during the demodulation process, we chose to demonstrate 15 micro-rings so that each one could operate in the near optimum condition. Future works on increasing the sensor number can focus on either reducing the separation of the resonant frequencies or broadening the bandwidth of the DOFC. The prior one includes the efforts to further increase the quality factor of the micro-ring sensors. For example, if micro-rings with quality factors up to $2 \times 10^6$ can be routinely fabricated, the full width at half maximum of the resonant frequency is about 100 MHz. Considering a ±100 MHz range for acoustic pressure induced frequency shifting, each micro-ring should at least occupy a frequency bandwidth of 200 MHz in the spectrum. Thus, a 40-GHz detection bandwidth can at most accommodate 200 micro-ring sensors in theory. As for the latter one, one may consider integrating both digital and optical approaches to generate ultrafine OFC with broad bandwidth. With a greater number of micro-rings that can be interrogated parallelly, two-dimensional sensor arrays, including ring shape or bowl shape, are possible to be realized and demonstrated for PAT, which is beneficial to the accommodation of various application scenarios. Moreover, two-dimensional sensor arrays enable the possibility to borrow the advanced concept of active illumination in optics with improved imaging quality[42,43]. Besides the perspective of design, the practical challenge of parallel interrogation also comes from the energy loss of the sensor array. To employ the same interrogation means for the two-dimensional sensor array, a bus waveguide needs to be designed with a relatively long zigzag path to couple all micro-rings distributed in the two-dimensional plane. Given inevitable coupling loss and a 0.2-dB/cm energy loss of the bus waveguide based on the current fabrication technique, these micro-ring sensors will operate in considerably different conditions. The demodulation speed of the DOFC is another important issue that needs to be considered for real-time imaging. Currently, it took us 0.05 s to demodulate an acoustic signal with a length of 10 μs (large enough to cover the field of view) for the sensor array with 15 elements. All signal processing was accomplished on a personal computer with Intel(R) Core (TM) i7-10700 CPU @ 2.90 GHz and 32 GB RAM. This demodulation speed can catch up with the laser repetition rate of real-time PAT (10 Hz). To incorporate more elements in the future, we may need to optimize the demodulation codes and upgrade the computational facility.

Moreover, it should be noted that the comb tooth in this study (39.0625 MHz) was chosen based on the quality factors of these micro-ring sensors. For quality factors within the range of $5 \sim 7 \times 10^5$, the full widths at half maximum of these resonant frequencies are typically 277-386 MHz, meaning that 7-10 sampling points were adopted to locate the precise location of the resonant frequency. In the ideal case, one comb tooth can be used to sample one resonant frequency theoretically. However, during experiments, more comb teeth, i.e., more sampling points, are needed to account for one resonant frequency. Such a choice is because the constantly shifting resonant frequency was not always located right at the position of the comb tooth during experiments. To accurately determine the position of the resonant frequency, one had to collect the values of all the sampling points and fit them to a Lorentzian-shaped curve. The position of the fitting curve with the minimum value then became the position of the resonant frequency. In practice, for micro-ring sensors with small quality factors, a sparser frequency comb could be used, which reduces the number of sampling points and alleviates the computational burden. In contrast, micro-ring sensors with high-quality factors generally require a finer frequency comb to locate the positions of the resonant frequencies with high accuracy.

In conclusion, we reported a micro-ring sensor array that contains 15 elements using chalcogenide glass. For such a new type of material that has rarely been explored in imaging and sensing, its large elastic-optical coefficients and low Young's modulus make it a promising candidate for optical ultrasound sensors. Specifically, the characterized parameters of this optical ultrasound array, including frequency bandwidth (175 MHz for −6 dB), NEP (2.2 mPaHz$^{-1/2}$), and acceptance angle (±30 degrees), are comparable with those of state-of-the-art optical ultrasound sensors being reported[19,27]. A comprehensive comparison is provided in Supplementary Note 4. We also developed an effective means of parallel interrogation of the micro-ring sensor array using only one source-detector pair. In contrast to previously demonstrated micro-ring sensor arrays that typically require one source-detector pair per channel[27], the developed means of parallel interrogation can greatly simplify the system setup while speeding up the data acquisition process. Such a simplification is particularly valuable for developing head-mount imaging devices with the optical ultrasound sensor array, where both compact size and fast data acquisition are required. Moreover, the means of parallel interrogation eliminates the necessity to synchronize the signals measured by each element, which benefits the processes of data collection and image reconstruction for PAT. With these advantages, demonstrations of PAT including imaging both fast-moving objects and live zebrafish at different growth stages were also provided. This work, which presents a chalcogenide-based micro-ring sensor array and its compatible DOFC-enabled parallel interrogation, is an important

milestone in advancing optical ultrasound sensor arrays toward both preclinical and clinical applications in PAT.

## Methods

### Structure of the sensor array

The main parts of the sensor array are the micro-ring sensors and the bus waveguide. The scanning electron micrographs of a micro-ring sensor and a bus waveguide are shown in Figs. 5(a) and 5(b), respectively. Each micro-ring sensor has a diameter of 40 μm, a cross-sectional height of 850 nm, and a width of 2.4 μm. They are also separated with a center-to-center distance of 400 μm and each of them occupies an effective area of about $0.85 \times 40 \times 40$ μm$^3$. Note that the resonant frequency of each micro-ring sensor was tuned to be slightly different through a photosensitive effect. The width and the height of the bus waveguide are 650 nm and 850 nm, respectively. Both the micro-ring sensor and the bus waveguide were fabricated using chalcogenide material with a composition of $Ge_{25}Sb_{10}S_{65}$, exhibiting a refractive index of 2.33 around 1550 nm. This material has a good photoelastic property with Young's modulus of 31.9 GPa and photoelastic coefficient of about 0.238, leading to a good sensitivity to ultrasound. These nano-fabricated structures sit on top of a silica ($SiO_2$) substrate and are covered with a 3-μm-thick polydimethylsiloxane (PDMS) cladding. This cladding is important in protecting micro-rings in an aqueous environment. No other micro/nanostructures are required to enhance the performance of the sensor array. A previous study showed that, in the architecture of silicon photonics, deformation in the PDMS cladding can contribute to the measured signals[19]. In Supplementary Note 5, we tested this effect with different types of cladding materials and found that the measured signals in this work are mainly contributed by chalcogenide-based micro-ring structures rather than PDMS claddings. By using the finite element method (COMSOL Multiphysics 5.6), Fig. 5(c) and (d) show the numerically simulated intensity distribution of the propagating modes inside the micro-

ring sensor and the waveguide, respectively, exhibiting good mode confinement. For practical considerations, we fabricated many micro-ring sensor arrays with different parameters on one chip, as shown in the microscope image in Fig. 2(e). After examining the performance, the part that only contains one sensor array was cleaved and encapsulated, occupying a small footprint of about 6 mm × 2 mm. As a comparison, a nickel coin with a 2-cm diameter was placed underneath, as shown in Fig. 5(f). In particular, the end of the bus waveguide, which is enclosed in a red box in Fig. 5(f), was attached to a single-mode optical fiber, with an enlarged view shown in Fig. 5(g). Currently, the fiber-to-waveguide insertion loss was estimated at around 6 dB for each side, indicating a total 12-dB insertion loss for the entire sensor chip. Detailed procedures regarding fiber-to-chip bonding can be found in Supplementary Note 6.

### The fabrication process of the sensor array

We first prepared an 850-nm-thick $Ge_{25}Sb_{10}S_{65}$ film and deposited it on a silicon wafer with a 3-μm-thick oxidized layer using the thermal evaporation technique. To prevent oxidation of the chalcogenide film during subsequent processes, 2-nm-thick alumina oxide was deposited through atomic layer deposition (Kemicro, ALD-100A). After spin-coating a layer of positive photoresist (Allresist, ARP6200) onto the chalcogenide film, electron beam lithography (Raith, EBPG5000+) was employed to write the pattern, i.e., the micro-rings and coupling waveguides. The pattern was then transferred onto the chalcogenide film by using reactive ion etching (Oxford Instrument, PlasmaPro 100RIE). The residual photoresist was removed through appropriate treatment with oxygen plasma, forming the basic structure of the on-chip micro-ring sensor array. In order to improve the durability of the chip, we spin-coated 4-μm-thick PDMS (Dow-corning, GZJ-184) to the entire chip for encapsulation. After completing the thermal curing of the PDMS film (curing agent ratio: 23%), the sensor array can now be exposed to the normal environment.

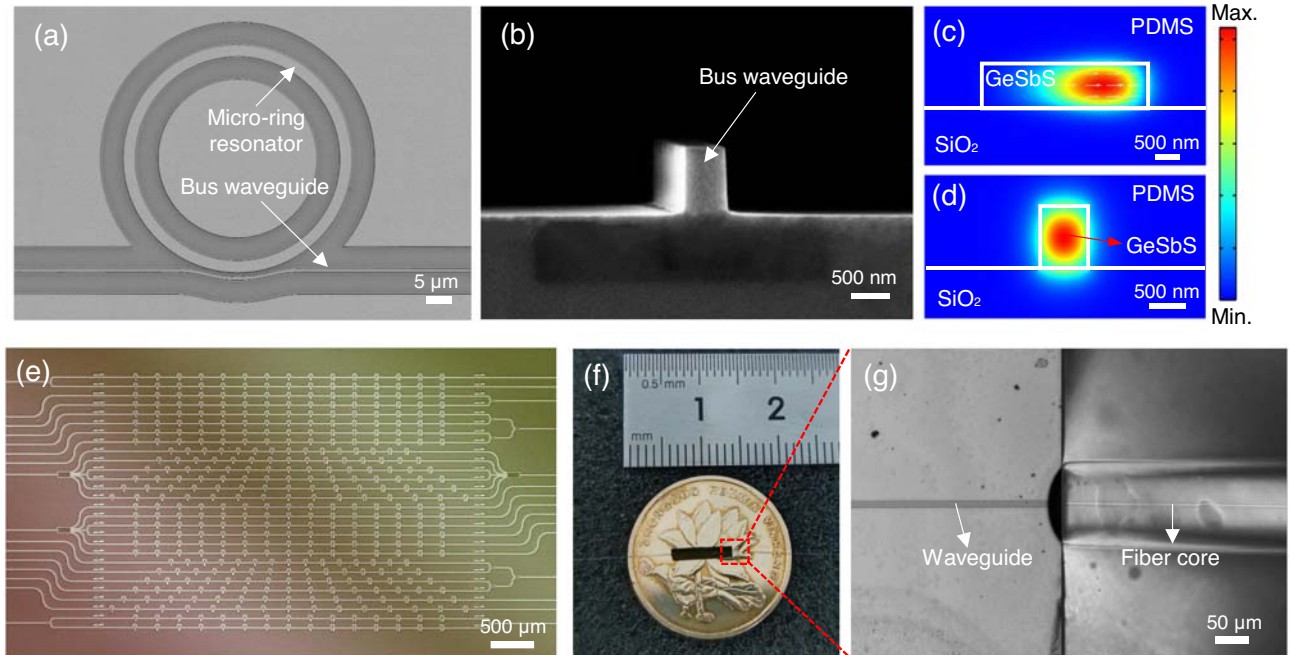

**Fig. 5 | Structural description of the chalcogenide-based micro-ring sensor array.** Each image has a different scale bar, which is provided in the lower right corner in white. **a, b** The scanning electron micrographs of a micro-ring sensor and a bus waveguide. **c, d** Numerical simulations of the mode profile, which are confined in the micro-ring sensor and the bus waveguide. **e** A microscope image of the chip, containing many micro-ring sensor arrays with different parameters. **f** A photo of the cleaved chip after encapsulation, which contains only one sensor array with a small footprint of about 6 mm × 2 mm. A nickel coin with a 2-cm diameter was placed underneath for comparison. **g** An enlarged view of the connection part between the bus waveguide and the single-mode optical fiber.

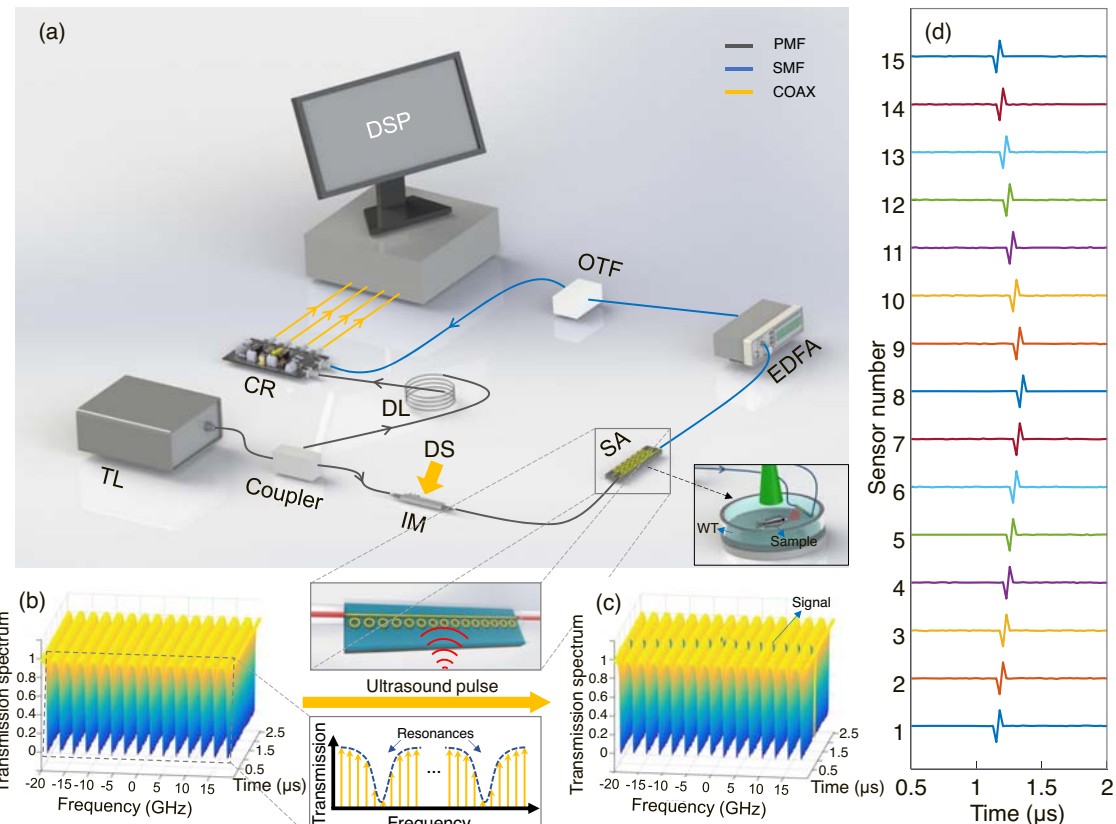

**Fig. 6 | Experimental setup and operational principle to perform photo-acoustic tomography using the chalcogenide-based micro-ring sensor array.**
**a** Experimental setup of the imaging system. The two insets show the generation and receiving processes of the PA signal. TL tunable laser, IM intensity modulator, DS digital signal, SA sensor array, WT water tank, EDFA erbium-doped fiber amplifier, OTF optical tunable filter, DL optical delay line, CR coherent receiver, DSP digital signal processing, PMF polarization maintaining fiber, SMF single mode fiber, COAX coaxial cable. **b** The measured transmission spectrum of the sensor array as a function of time in the null case. Inset: a schematic diagram that illustrates how the DOFC samples the transmission spectrum of the sensor array. DOFC digital optical frequency comb. **c** The measured transmission spectrum of the sensor array when ultrasound is present to modulate the sensor array. **d** The reconstructed ultrasound signal as a function of time for each micro-ring sensor.

## Experimental setup for PAT using the sensor array

The experimental setup of employing the micro-ring sensor array for PAT is shown in Fig. 6(a). A continuous-wave narrow-linewidth tunable laser (Keysight, 8164B, 10-kHz linewidth, 1550 nm) was chosen as the optical source for the sensor array. Its frequency was tuned to be close to the resonant frequencies of these micro-rings and its output power was set to 10 mW. The output light of the laser was divided into a signal beam and a local oscillator through a polarization-maintaining fiber coupler (Thorlabs, PNH1550R5A1). The signal beam (~0.12 mW in the bus waveguide) interacted with the sensor array to probe the information carried by the ultrasonic wave. In this work, we employed a DOFC to realize parallel interrogation of the sensor array, which has the advantages of stability, flexibility, and tunability. Pictorially, the DOFC exhibits a comb structure with a fine comb tooth in the frequency domain, which can be measured through a single coherent detector in the time domain. By quantifying the changes in amplitude and phase of these comb tooths, the transmission spectrum of the sensor array in the frequency domain can be accurately determined. By scrutinizing the temporal change in the transmission spectrum, time-dependent acoustic signals can be determined. It is worth noting that the DOFC has been demonstrated previously for ultrafine spectral measurement (0.01 pm resolution)[44] and low-frequency ultrasound detection (165 kHz)[45]. For a fiber-based resonator, a similar comb structure was also created optically with pulsed light and two fiber Bragg gratings for ultrasonic detection[12]. In this work, the interrogation system mainly contains two parts, i.e., the

generation and demodulation processes of the DOFC. The generation process started with a digital signal with a multi-carrier bandwidth of 40 GHz and a sequence length of 1536, which was synthesized through the orthogonal frequency division multiplexing method. The digital signal was then fed into an arbitrary waveform generator (Keysight, M8195A) with a sampling frequency of 60 GHz. In this condition, the corresponding spacing of the subcarrier, i.e., the comb tooth spacing, is 39.0625 MHz (60 GHz/1536). The signal was then amplified and periodically sent to an intensity modulator (Xblue, MXER-LM-20). By setting the intensity modulator with a bias control at the node point, a carrier-suppressed double-sideband modulated DOFC signal was generated with a bandwidth of about 40 GHz (320 pm @1550 nm). Detailed mathematical descriptions of the generation of the DOFC can be found in Supplementary Note 7. The DOFC is then passed through the sensor array to probe PA signals. Having acquired the information from the sensor array, the DOFC was then post-amplified by an erbium-doped fiber amplifier (Amonics, AEDFA-PA-35-B-FA) and filtered by an optical tunable filter (Santec, OTF-350). Finally, the DOFC was combined with the local oscillator, which was subsequently measured by a coherent receiver (Finisar, CPRV122xA) and digitized by an oscilloscope (Lecroy, 10-36Zi-A). To avoid the phase noise of the laser and increase measurement accuracy, a delay line was introduced to match the path length difference. To demodulate the DOFC, the measured signal was Fourier transformed into the frequency domain, leading to the transmission spectrum of the sensor array (also detailed in

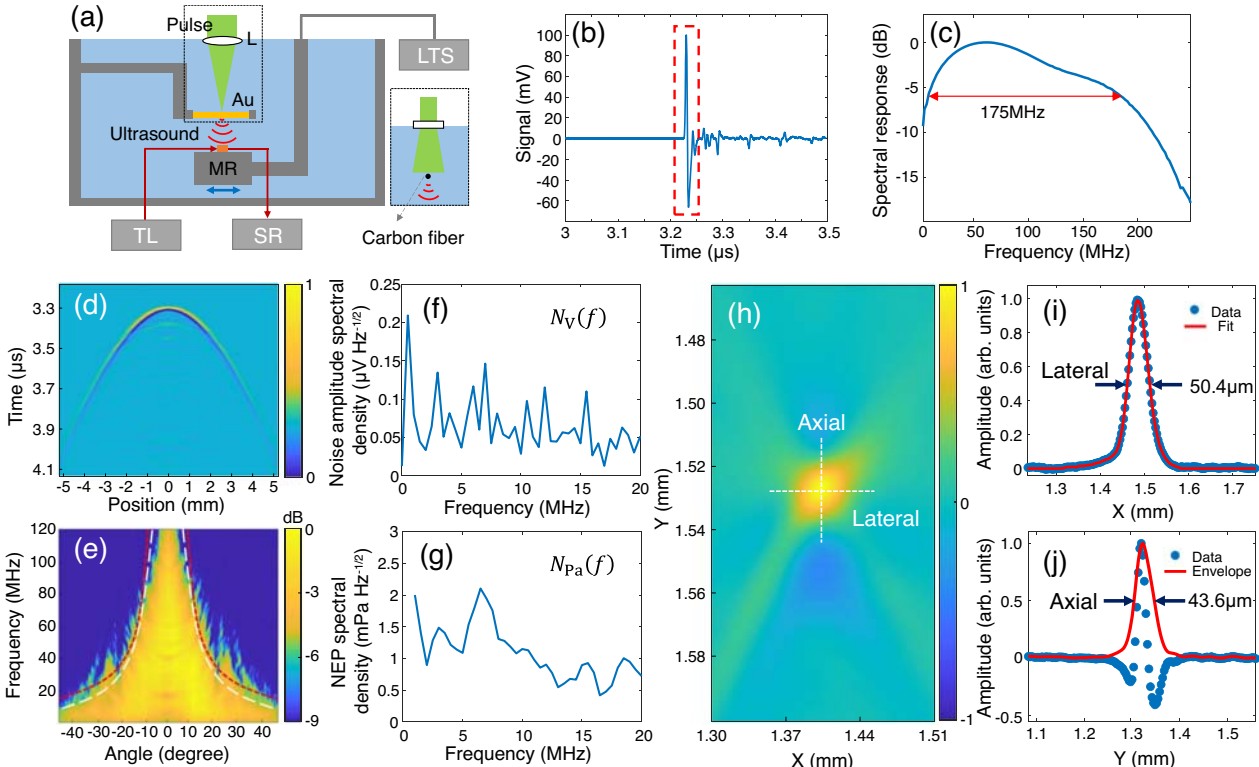

**Fig. 7 | Characterization of the micro-ring ultrasound sensor. a** Schematic illustration of the characterization system that focuses pulsed light onto a golden thin film to generate a point-like ultrasound source. L lens, TL tunable laser, SR signal receiver, MR micro-ring. LTS linear translational stage. **b** The measured PA signal as a function of time. A time window (red dashed box) was implemented to keep only the first arriving signals while excluding the following ones due to multiple reflections. **c** The frequency response of the micro-ring sensor. The center frequency is around 60 MHz, and the −6 dB bandwidth is about 175 MHz. **d** Amplitude map of the measured PA signal as a function of time and translational distance. **e** Frequency response as a function of the acceptance angle. Two experimentally achieved −3 dB lines are also provided in white, matching the theoretical estimations in red. **f** Noise amplitude spectral density of the micro-ring sensor. **g** Noise-equivalent-pressure (NEP) spectral density of the micro-ring sensor. **h** Reconstructed images of the cross-section of the carbon fiber buried inside the agar. **i** One-dimensional profile along the lateral direction, suggesting a lateral resolution of 50.4 μm. **j** One-dimensional profile along the axial direction, suggesting an axial resolution of 43.6 μm.

Supplementary Note 7). It is worth emphasizing that the employment of the DOFC allows the determination of the transmission spectrum in a one-time measurement, without the need for time-consuming frequency sweeping. We also note that the employment of the DOFC does not reduce the amount of data required for image reconstruction. In this condition, the coherent receiver is expected to have a much larger bandwidth than a standard photodiode. Moreover, extracting the entire transmission spectrum through the DOFC lowers the sampling rate, which is a compromise to the detecting bandwidth at $\Delta f/2$ ($\Delta f$ is the comb tooth spacing) and is detailed in Supplementary Note 4. As a result, the developed means of parallel interrogation essentially trades the bandwidth of the detection for the compactness of the imaging system. In the measured transmission spectrum, the amount of resonant frequency shift reflects the amplitude of the received PA signal. Taking 15 micro-rings as an example, Fig. 6(b) plots the measured transmission spectrum as a function of time in the null case, which is measured by the DOFC. Notably, the resonant frequency of each micro-ring remains as a flat line, indicating a stable detecting environment due to the encapsulation of the sensor array. When the laser-induced ultrasound signal is present to modulate these micro-rings, Fig. 6(c) plots the measured transmission spectrum as a function of time. In this condition, the amount of resonant frequency shift for each micro-ring sensor faithfully represents the amplitude of the PA signal measured by each element. The time delay reflects the relative distance between the ultrasonic source and the micro-ring sensor. As a result, the reconstructed PA signal as a function of time for each micro-ring sensor is illustrated in Fig. 6(d).

## Characterization of the micro-ring sensor

We first characterize the performance of the micro-ring sensor in an aqueous environment. As shown in Fig. 7(a), a point ultrasound source was generated by focusing pulsed light (Elforlight, SPOT-10-200-532, 2.6-ns pulse width) onto a 200-nm-thick golden thin film (fabrication procedure described in Supplementary Note 8)[27,30,46]. The diameter of the focal spot was about 9 μm. The micro-ring sensor array was positioned 5 mm aside to measure the PA signal. A motorized linear translational stage was used to control the position of the sensor array. Both the thin film and the micro-ring sensor were immersed inside the water which serves as the coupling medium for the ultrasonic wave. For fair assessment, we describe the characterization process for the micro-ring sensor with the 8th largest quality factor in detail. For this typical micro-ring sensor, the received PA signal as a function of time is plotted in Fig. 7(b), which can be treated as the impulse response of delta excitation. To remove unwanted signals due to multiple reflections between substrates, we enforced a time window to keep only the first arriving signals (highlighted in the red dashed box). Thus, the frequency response of the micro-ring sensor can be estimated by taking the Fourier transformation to the time-gated signal, which is plotted in Fig. 7(c). The central frequency locates around 60 MHz, and the −6 dB bandwidth is estimated to be 175 MHz (the -3 dB bandwidth is 115 MHz). We also quantified the acceptance angle of the micro-ring

sensor by continuously scanning the position of the sensor, while keeping the position of the point ultrasound source. Figure 7(d) shows the amplitude map of the measured PA signal as a function of time and translational distance. By transforming time and position into frequency and acceptance angle, the frequency response of the micro-ring sensor as a function of the acceptance angle is shown in Fig. 7(e) (one-dimensional profiles of detailed time and frequency responses are provided in Supplementary Note 9). Moreover, experimentally achieved -3-dB lines are labeled in white, showing a similar structure and trend to the theoretically estimated ones in red (detailed in Supplementary Note 9 using the procedure described in Ref. 47). These results indicate that for frequencies up to around 25 MHz, wide acceptance angles covering about ±30 degrees can be achieved. Then, we followed the procedures described in Ref. 27 to characterize the sensitivity of the micro-ring sensor, with the assistance of a calibrated hydrophone (Precision Acoustics, NH0200, 20-MHz bandwidth). The noise amplitude spectral density was calculated when no ultrasound was present, which is provided in Fig. 7(f). By dividing the noise amplitude spectral density with respect to the sensitivity, Fig. 7(g) shows that the NEP spectral density is below 2.2 mPaHz$^{-1/2}$, leading to the measured NEP of 7.1 Pa within 20-MHz bandwidth. All other micro-ring sensors were characterized in the same way, exhibiting NEPs within ±10% of the value reported above. These values are comparable to those of state-of-the-art optical ultrasound sensors[19,27]. Detailed calculation processes for obtaining these values can be found in Supplementary Note 10. It should be noted that the use of the parallel interrogation method introduces extra noise and thus increases the NEP. For the same micro-ring sensor, the NEP spectral density and the NEP increase to 13.9 mPaHz$^{-1/2}$ and 36.9 Pa within a 20-MHz bandwidth, respectively (Detailed in Supplementary Note 11). The NEPs for the rest of the micro-ring sensors are within the ±10% range. Compared to the one characterized above, the increased NEPs here are due to the evenly distributed power into each comb tooth (1/1536 theoretically), which is a compromise to enable parallel interrogation. Nonetheless, these values are still comparable to the NEP reported with conventional methods in Ref. 10 (24 Pa within a 10-MHz bandwidth). Moreover, the loss in sensitivity going from a single detector to an array is not large when compared to other techniques. For example, it was shown in Ref. 48 that using pulse transmission amplitude monitoring for parallel readout of arrays could cause orders of magnitude loss in sensitivity compared to the single-element approach. In PAT, the imaging resolution is determined by the detecting bandwidth of the micro-ring sensor. This parameter was quantified by replacing the golden thin film in Fig. 7(a) with a horizontally placed carbon fiber (6 μm in diameter) buried inside the agar. The cross-section of the fiber was imaged to provide information on the point spread function of the imaging system. The micro-ring sensor was linearly translated with a step size of 0.7 μm with a 2.8-mm range. UBP[40] was employed to synthesize detected PA signals at each position and produce the image in Fig. 7(h). One-dimensional profiles for quantifying both the lateral and axial resolutions are illustrated in Fig. 7(i) and (j), and their envelopes exhibiting full width at half maximum of about 50.4 μm and 43.6 μm, respectively. These curves have similar shapes to the ones reported in the literature[19,34,49,50] and the ones simulated numerically (detailed in Supplementary Note 12). Theoretically, for partial view PAT, the lateral resolution is given by $0.71\,v/(NAf_0) \approx 35.5$ μm[51]. Here, $v = 1{,}500$ mm/ms is the speed of sound, NA $\approx \sin(30°) = 0.5$ is the numerical aperture estimated using the acceptance angle of the micro-ring sensor, and $f_0 = 60$ MHz is the central frequency. The small discrepancy between the estimated value (35.5 μm) and the measured one (50.4 μm) is

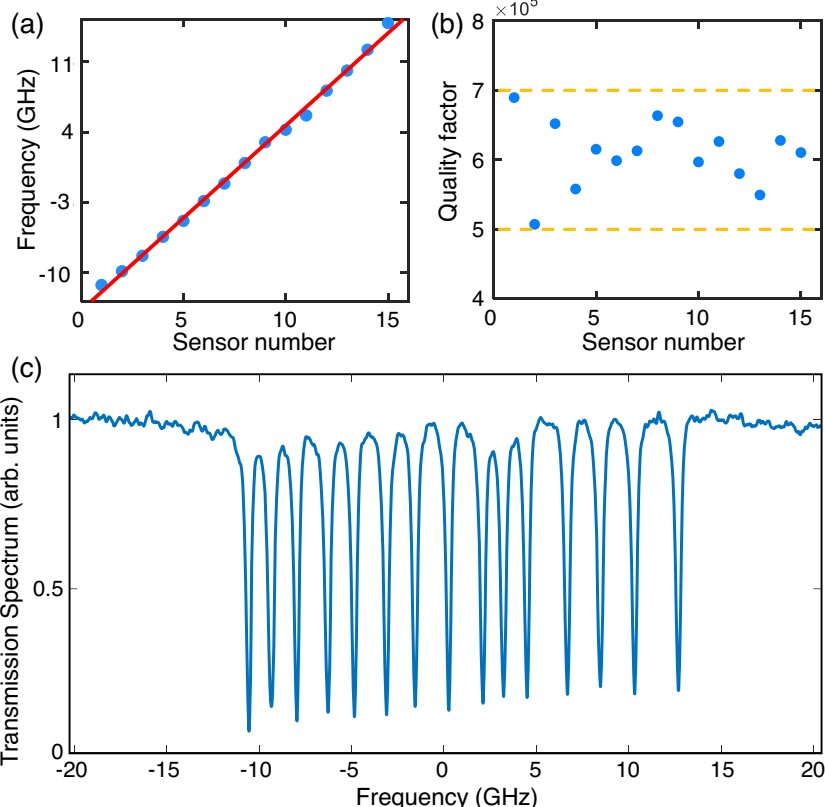

**Fig. 8 | Frequency tuning for the micro-ring sensor array with 15 micro-rings. a** Resonant frequencies of the micro-ring sensors after frequency tuning, which are ordered and equally spaced with respect to their labeling. **b** The quality factors of these micro-ring sensors after frequency tuning, exhibiting good consistency. **c** A typical transmission spectrum of the micro-ring sensor array after frequency tuning, exhibiting 15 discrete resonant dips.

because $f_0 = 60$ MHz applies for only small acceptance angles so that the average central frequency over the entire angular range is smaller. Moreover, the theoretically estimated axial resolution is given by $0.88\, v/\Delta f \approx 11.5\,\mu$m, where $\Delta f = 115$ MHz is the 3-dB bandwidth[51]. This value (11.5 μm) is also smaller than the measured one (43.6 μm). This observation originates from the fact that the bandwidth reduces considerably for large acceptance angles so that the average effective bandwidth over the entire angular range is much smaller. Moreover, the fact that high-frequency components attenuate much more than low-frequency counterparts in agar that covers the carbon fiber might also contribute.

### Frequency tuning of the micro-ring sensor array

The micro-ring sensor array used in this work contains 15 micro-rings, serving as a linear array for PAT. Since slight deviations in terms of the diameter of these micro-rings (10 nm) are inevitable due to the fabrication precision of electron beam lithography, the resonant frequencies of these micro-rings were randomly displayed at the beginning. To facilitate the demodulation process of the DOFC and enable parallel interrogation, ordering and equally spacing these resonant frequencies are critically important. This requirement can be fulfilled by exploiting the strong photosensitive effect of chalcogenide glasses[52,53]. Experimentally, we used an optical fiber to illuminate each micro-ring sensor with 532-nm pulsed light. By controlling the illuminating intensity and time duration, the resonant frequency of the micro-ring sensor is reconfigurable with the desired amount. Illuminating light can be removed later after reconfiguring these resonant frequencies. Detailed operational procedures for rearranging the resonant frequencies of these micro-rings can be found in Supplementary Note 13. After the tuning process, it is shown in Fig. 8(a) that the resonant frequencies of these micro-ring sensors are now ordered with respect to their labeling and equally spaced. The average separation between adjacent resonance frequencies is about 1.66 GHz (0.013 nm @ 1550 nm), and these resonant frequencies occupy an overall spectrum range of 24.9 GHz (0.199 nm @ 1550 nm). As shown in Fig. 8(b), after frequency tuning, the quality factors of these micro-ring sensors were also quantified with good consistency, which fell within the range of 5 to $7 \times 10^5$. As a result, a typical transmission spectrum of the micro-ring sensor array is shown in Fig. 8(c), exhibiting 15 distinct and equally separated resonant dips within a frequency range of −11.45-13.45 GHz (with respect to the center of the DOFC). These resonant dips can stay well resolved within 6 h, which is detailed in Supplementary Note 14.

### Reporting summary

Further information on research design is available in the Nature Portfolio Reporting Summary linked to this article.

## Data availability

The imaging data generated in this study have been deposited in the Zenodo database under accession code https://doi.org/10.5281/zenodo.7688935.

## Code availability

Codes of DOFC-enabled interrogation means for retrieving PA signals from the micro-ring sensor array have been deposited in the Zenodo database under accession code https://doi.org/10.5281/zenodo.7481865.

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

## Acknowledgements

Y.S. thank Prof. Biqin Dong at Fudan University for theoretical estimations on the angular response of micro-ring sensors, Prof. Xiaohua Feng at Zhejiang Laboratory for discussions on reconstruction algorithms, Prof. Long Jin and Prof. Jun Ma at Jinan University for discussions on imaging with planar geometries. This work was supported in part by the National Key Research and Development Program of China (2019YFA0706301), the National Natural Science Foundation of China (U2001601, 62105379, 12004446, 92150102), the China Postdoctoral Science Foundation (2021M693598), the Innovation Group Project of Southern Marine Science and Engineering Guangdong Laboratory (Zhuhai) (No. SML2022007), and the Key-Area Research and Development Program of Guangdong Province (2020B0101080002), and the Marine Economy Development Special Fund (Six Marine Industries) under Department of Natural Resources of Guangdong Province (GDNRC [2021]33).

## Author contributions

J.P. built the DOFC system and analyzed the data. Q.L. built the photoacoustic imaging system and analyzed the data. Y.F. and R.Z. helped with data analysis. Z.F. fabricated on-chip devices. S.Y. prepared of chalcogenide films. W.S. packed the devices. B.Z., Q.S., and J.C. assisted in the experimental design and data interpretation. Y.S. and Z.L. provided overall supervision.

## Competing interests

The authors declare no competing interests.
