## [Peer Review File · Nature Communications]

REVIEWER COMMENTS

Reviewer #1 (Remarks to the Author):

In their work, the authors demonstrate a linear array of ultrasound detector in which the individual elements are microrings fabricated in chalcogenide glass. The authors report high sensitivity and bandwidth and demonstrate their system for imaging a leaf and a zebra fish. The work is technically sound and the challenge of developing arrays of optical detectors is very important to the fields of ultrasound and photoacoustics. However, there are some issues with the novelty of the work and the way it is presented. I believe major modifications need to be made to the manuscript before it could be considered for publication in Nature Communications:

1) It is not clear what the main novelty of the work. The type of resolutions reported in the work have been achieved by the UCL group using Fabry-Perots, where parallel interrogation was reported for up to 16 channels. PDMS-coated resonators have been reported in silicon by Hazan et al (Ref. 19) and the frequency-comb interrogation scheme has already been published. The authors need to better pinpoint the innovation of the technology, in terms unique performance and/or some missing component that didn't exist previously that allowed them to combine all these techniques into a single system.

2) The authors claim that using a single coherent detector is a much better option than using multiple photodiodes, as done in other groups. This argument has two flaws. First, photodiodes are cheap, and it is not obvious to me that a single coherent detector is cheaper than 10 photodiodes. Second, the true cost is not in the photodiodes, which are cheap components, but in the sampling electronics. The authors fail to mention that the sampling BW of their system must be much higher than the acoustic BW since they measure the interference between all the resonators. The cost and complexity of sampling systems is not just in the number of channels, but also in the total bit/s data rate, which would be higher in the proposed scheme because not all the BW is utilized.

3) The method strongly relies on tuning the microrings' spectrum to fit that of the frequency comb. However, it is well known that there could be drifts in photosensitivity-induced structures over time. In addition, temperature variations could also scramble the microring spectrum. Could the author remeasure the spectrum of the micro-ring array now and show if any differences have been observed? Also, the authors should perform a temperature-dependent measurement and comment on how stable the micro-ring comb spectrum is over a normal temperature range (20-30 Celsius).

4) More information should be given on the coherence detection scheme to explain how the raw signal from a single coherent detector is transformed into an array. This should include showing the signals at each step of the process.

5) The authors claim that the PDMS layer was used only to protect the micro-ring and that the signal is a result of the high photo-elastic coefficient of the glass. However, this contradicts the conclusion of Hazan et al. in silicon, where it was experimentally shown that the signal actually comes from the PDMS. The authors should substantiate their claim by using harder coating materials (e.g. normal silica, or

PMMA) and show that the response does not change. If the coating determines the signal characteristics, it is possible that the authors were wrong about their conclusion and would need to modify the manuscript accordingly. In any case, the comparison to the alternative coating material is essential. If PDMS is essential for the performance obtained in this paper, the work of Hazan et al. should be properly cited.

6) The authors claim a 30-degree acceptance angle, referring to Fig. 4e. It would be good to elaborate on that. Could the authors show that time- and frequency-domain responses obtained at several angles (e.g. 10, 20, etc.). It would be good to see what happens beyond 30 degrees. The experimental results should be justified with a numerical simulation. The theoretical angular response of a microring is can be found in "Dong, B., Sun, C. & Zhang, H. F. Optical detection of ultrasound in photoacoustic imaging. *IEEE Trans. Biomed. Eng.* 64, 4–15 (2017)."

7) The authors should generally explain why their resolution is not higher (Fig. 4). Generally speaking, a 40-micron microring should give a 40-micro lateral resolution. Why is that not the case here? Also the axial resolution is lower than what I would expect for a 175 MHz bandwidth.

8) The PSF shown in Fig. 4h is not so clear and does not look like the theoretical prediction (<https://doi.org/10.1364/AO.47.000561> -Fig. 9), which has also been verified by Hazan et al.

9) Using a rotation-based imaging system defies the logic of using optical sensors. As has been shown the UCL group, optical sensors have the advantage of working in a planar geometry. The authors need to show that their technique can do the same and still get good images, even if some resolution is lost.

10) In Figs. 7a-d, it is not clear what the detection geometry was. Where the detector in-plane or arranged vertically. It feels like the benefit of using more detectors was not just from SNR, but also from having more projections. I suspect that the 1-sensor measurement would still not be as good as the one from the array even if the authors averaged. A fair comparison with the same number of signals is needed to answer this question.

11) It is not clear what the imaging geometry of the zebrafish was.

12) What was the fiber-to-fiber insertion loss of the system? Could the authors elaborate on their fiber-to-chip bonding procedure?

13) Could the authors provide more information on how the black leaf was produced? Namely, what was the ink, was it diluted, and what was the infusion protocol? A photo of the leaf would be beneficial.

14) What was the fabrication procedure of the gold layer? What type of machine was used to deposit the gold layer and how long did it take? The authors could use the following paper as reference: Seeger, M., Soliman, D., Aguirre, J. et al. Pushing the boundaries of optoacoustic microscopy by total impulse response characterization. *Nat Commun* 11, 2910 (2020).

15) What was approximately the diameter of the focused optical beam used to generate the optoacoustic point source?

16) What was the laser power, and how much was coupled into the waveguide?

Reviewer #2 (Remarks to the Author):

This work has made two major contributions to optical detection of photoacoustics. First the authors demonstrated 15-microring sensor array made of chalcogenide glass for PAT imaging. The sensors show high quality factor (10^5), high bandwidth (175 MHz) and low NEP ($2.2 \text{ mPa}\cdot\text{Hz}^{-1}$). Second, this work synthesizes digital optical frequency comb to interrogate the sensor array in parallel fashion with only one source and sensor. This method greatly simplifies the interrogation method and enables high speed photoacoustic tomography with microring array sensor. Photoacoustic imaging of fast-moving object, leaf veins and live zebra fish have been demonstrated with the proposed method.

1. Overall, this is an excellent work. A few questions are listed below for authors' clarification:
2. In line 122, what is the photoelastic coefficient of the material? Only young's modulus of 31.9 GPa is given.
3. The signal comes primarily from refractive index change of the microring (photoelastic effect), right? does the elastic deformation of the PDMS coating contribute to the detection?
4. In figure 2(b), is the one in the center the bus waveguide and are the 2 on the edges the microring? Why does the cross-section profile look like this? It is better to indicate the region on figure 2(a).
5. DOFC is described mathematically in the SI. It will be helpful to readers if the authors can provide a short intuitive understanding of its working principle in the main text. This is one of the most important contributions of the work.
6. In the section of "Characterization of the micro-ring sensor", it mentions the NEP is 7.1 Pa within 20-MHz. Is it an averaged value of the 15 sensors of the array? What is the uniformity of the sensitivity and the NEP of the 15 sensors.
7. It mentions that the NEP of the sensor is comparable to the state-of-the-art ultrasound sensor. How about the parallel interrogation method compared with conventional method that is used in reference [10]? Does the parallel interrogation method introduce extra noise and change the NEP?
8. Since the parallel interrogation is a relatively new method. It will be helpful to include some benefits/ challenges of this method.

Reviewer #3 (Remarks to the Author):

In this manuscript, Pan et al. report the fabrication of optical ultrasound sensors using chalcogenide compound materials. In particular, a sensor array that contains 15 microrings was demonstrated for photoacoustic tomography. The parameters of these microring sensors, such as sensitivity and bandwidth, are attractive to the community of photoacoustic imaging. Moreover, to enable parallel interrogation by using all these elements simultaneously, the authors further introduced a new and novel means by developing a digital optical frequency comb with an ultrafine comb tooth. With parallel interrogation, photoacoustic imaging for fast-moving objects and live animals was demonstrated, confirming the validity of this work. In general, this is thorough and valid work. Given the means of parallel interrogation of optical ultrasound sensor array was rarely reported and in urgent need for the field of optical ultrasound sensor enabled photoacoustic imaging, this paper represents an important step forward. The paper is well written and fairly complete in terms of data. Still, I have a few questions

1. The sensitivity of the micro-ring sensor was quantified to have a very small NEP down to 7.1 Pa. Although this value seems to be attractive, however, it seems to be the property of the optical ultrasound sensors only. When employing the DOFC-enabled parallel interrogation, will additional noise occur? If yes, please provide this situation in more detail, as it may become one of the limitations of this interrogation means. Generally speaking, using a more complex measurement scheme usually introduces more noise.
2. When quantifying the resolutions in Fig. 4, the image seems to be quite pixelated. The authors should spend more effort on determining these values more accurately. Also, the vertical axis of the -6-dB bandwidth seems to be mistakenly labeled. Please correct.
3. The experimentally achieved axial and lateral resolutions are 24 and 80 μm , respectively. These values seem to be inconsistent with the central frequency (60 MHz) and bandwidth (175 MHz) of the microring sensor. The authors should either provide detailed explanations on justifying this discrepancy or correct these values.
4. The comb tooth of the digital optical frequency comb is 39 MHz. The authors provide little information on the choice of this value. Could we use a larger or smaller comb tooth? What are the possible affections on the imaging performance?
5. I suggest the authors estimate the moving velocity of fast-moving objects, which demonstrates the capability of the micro-ring sensor array and its means of parallel interrogation means.
6. The resolution of live animals does not seem to be as good as tens of micrometers. This is natural as I believe high-frequency ultrasound may not propagate well in thick scattering samples or not even be fully excited in zebrafish. If this is the case, the authors should describe this issue. In other words, we

may not need such a high-frequency bandwidth for imaging thick biological samples. A detailed discussion on this issue may be helpful to researchers in the field to design their optical ultrasound sensors in the future.

7. What is the demodulation speed of the interrogation means? Can it reach 10 Hz to be consistent with the repetition rate of the pulse laser source in photoacoustic tomography?

8. Linear array may not be the optimal choice for single-shot photoacoustic tomography, as it suffers from the problem of limited view. This problem was alleviated by scanning the samples in the paper but at the cost of multiple shots. I suggest the authors consider the possibility to extend this sensor array into a two-dimensional structure. What are the possible challenges to making a two-dimensional sensor array? Can we use the same interrogation means with one coupling waveguide? What is the limiting factor that determines the maximum number of elements?

9. What is the stability of this sensor array? Since the authors use visible light to tune the resonant frequencies of each microring sensor, I suspect the sensor array may be susceptible to the environment, such as temperature, humidity, and ambient light illumination. The authors should quantify this effect and provide more details on this point.

10. From the perspective of design, would the authors be able to comment on the scalability of this device? In theory, what would be the maximum number of microrings based on the current configuration? Would it be possible to extend this technique to fabricate a 2D array?

In conclusion, the paper presents an impressive step forward, but its value to the community could be enhanced by clearer presenting data.

Point-by-point responses to reviewers' comments

Reviewer #1:

In their work, the authors demonstrate a linear array of ultrasound detector in which the individual elements are microrings fabricated in chalcogenide glass. The authors report high sensitivity and bandwidth and demonstrate their system for imaging a leaf and a zebra fish. The work is technically sound and the challenge of developing arrays of optical detectors is very important to the fields of ultrasound and photoacoustics. However, there are some issues with the novelty of the work and the way it is presented. I believe major modifications need to be made to the manuscript before it could be considered for publication in Nature Communications:

1. *It is not clear what the main novelty of the work. The type of resolutions reported in the work have been achieved by the UCL group using Fabry-Perots, where parallel interrogation was reported for up to 16 channels. PDMS-coated resonators have been reported in silicon by Hazan et al (Ref. 19) and the frequency-comb interrogation scheme has already been published. The authors need to better pinpoint the innovation of the technology, in terms unique performance and/or some missing component that didn't exist previously that allowed them to combine all these techniques into a single system.*

Response: We thank the reviewer for his suggestion to better pinpoint the innovation of our work. The main novelty of our work includes two parts. First, we demonstrated an on-chip chalcogenide-based micro-ring sensor array with 15 elements for photoacoustic tomography. These micro-ring sensors show high quality factors (10^5), large bandwidth (175 MHz), and low noise-equivalent pressure ($2.2 \text{ mPaHz}^{-1/2}$). These values are comparable to those of state-of-the-art optical ultrasound sensors. Second, we developed an interrogation means for the sensor array by synthesizing a digital optical frequency comb (DOFC), which simplifies the imaging setup by employing only one continuous-wave light source and one coherent photodetector. Since on-chip micro-rings with high-quality factors generally exhibit resonant dips with narrow linewidth, the unique property of the DOFC with a tunable and ultra-narrow comb tooth is well suited for measuring the transmission spectrum of the sensor array with high accuracy. The complete knowledge of the transmission spectrum allows the acoustic signals measured by all micro-ring sensors to be acquired simultaneously. Therefore, the marriage of these two technical innovations breeds a compact imaging system of photoacoustic tomography. To emphasize the novelty of our work, in the revised main text, we modified the description in the section “**Introduction**” and emphasize the innovation of this work:

a. *In contrast to optically generated frequency combs, the DOFC holds the unique advantage of generating an ultra-narrow and tunable comb tooth. Since on-chip micro-rings with high-quality factors generally exhibit resonant dips with narrow linewidth, this unique property of the DOFC is well suited to locate resonant frequencies for all micro-ring sensors in parallel with high*

accuracy.

- b. These results indicate that the marriage of these two technical innovations, i.e., high-performance micro-ring sensor array and DOFC-enabled parallel interrogation means, breeds a compact imaging system of PAT using an ultrasound optical sensor array, offering great prospects for clinical applications.

2. *The authors claim that using a single coherent detector is a much better option than using multiple photodiodes, as done in other groups. This argument has two flaws. First, photodiodes are cheap, and it is not obvious to me that a single coherent detector is cheaper than 10 photodiodes. Second, the true cost is not in the photodiodes, which are cheap components, but in the sampling electronics. The authors fail to mention that the sampling BW of their system must be much higher than the acoustic BW since they measure the interference between all the resonators. The cost and complexity of sampling systems is not just in the number of channels, but also in the total bit/s data rate, which would be higher in the proposed scheme because not all the BW is utilized.*

Response: We thank the reviewer for pointing this out. We did not claim the developed means of parallel interrogation used a cheaper device than the 10 photodiodes in the manuscript. The benefit of using a single coherent detector in terms of using multiple photodiodes is for the simplification and compactness of the imaging system. Nonetheless, we agree with the authors that we indeed used a much higher sampling bandwidth to realize parallel interrogation. This is natural as we need to collect more information (from all 15 micro-rings) within the same amount of time. To avoid confusing potential readers, we added the following sentences in the section “**Method**” of the revised main text to discuss this issue, which explicitly shows that the sampling bandwidth of our system is higher: *We also note that the employment of the DOFC does not reduce the amount of data required for image reconstruction. In this condition, the coherent receiver is expected to have a much larger bandwidth than a standard photodiode. Moreover, extracting the entire transmission spectrum further sacrifices the bandwidth to some extent. As a result, the developed means of parallel interrogation essentially trades the bandwidth of the detection for the compactness of the imaging system.*

3. *The method strongly relies on tuning the microrings’ spectrum to fit that of the frequency comb. However, it is well known that there could be drifts in photosensitivity-induced structures over time. In addition, temperature variations could also scramble the microring spectrum. Could the author remeasure the spectrum of the micro-ring array now and show if any differences have been observed? Also, the authors should perform a temperature-dependent measurement and comment on how stable the micro-ring comb spectrum is over a normal temperature range (20-30 Celsius).*

Response: We thank the reviewer for asking this important question. The spectrum of the sensor

array can be quite stable for several hours, which is sufficiently long to perform PAT. On one hand, we also found that temperature change in water causes resonant frequencies of all micro-ring sensors to shift in the same direction with roughly the same amount, indicating that temperature fluctuations in the water do not scramble the spectrum. On the other hand, we further found that temperature fluctuations in the room do not considerably affect the spectrum of the sensor array. The data that support the above claims, i.e., spectrum stability as a function of time and temperature variation, are supplemented in the revised submission. In particular, in the revised main text, we added the following sentence in the section “**Frequency tuning of the micro-ring sensor array**”: **These resonant dips can stay well resolved within 6 hours, which is detailed in Supplementary Note 13.**

In the revised Supplementary Information, we also added a new section “Supplementary Note 13” to describe the stability issue in terms of both time and temperature.

Supplementary Note 13. The examination of the spectrum stability of the micro-ring sensor array

The DOFC method strongly relies on the delicate tuning of the sensor spectrum. In this section, we examined the spectrum stability of the micro-ring sensor array. Firstly, we measured the transmission spectrum of the sensor array over time for 6 hours, which is shown in Supplementary Fig. 12. As we can see from the figure, the photo-sensitive effect of the material did cause observable resonant frequency shifts in the spectrum. Nonetheless, all 15 resonant dips are still well-resolvable, allowing sustainable imaging operation using the sensor array. For comparison purposes, grey dashed lines were used to denote the position of the original resonant frequencies. Quantitatively, the mean absolute frequency drifts of the 15 resonant frequencies after 1 hour, 2 hours, 3 hours, 4 hours, 5 hours, and 6 hours are 0.319 GHz, 0.451 GHz, 0.451 GHz, 0.312 GHz, 0.401 GHz, and 0.375 GHz, respectively. Correspondingly, the standard deviation of the frequency drifts of the 15 resonant frequencies after 1 hour, 2 hours, 3 hours, 4 hours, 5 hours, and 6 hours are 0.067 GHz, 0.267 GHz, 0.267 GHz, 0.188 GHz, 0.309 GHz, and 0.208 GHz, respectively. All these values are much smaller than the average separation of adjacent resonant dips (1.66 GHz). These results show that the micro-ring sensor array can still function normally even after being placed in the aqueous environment for 6 hours.

Supplementary Fig. 12. The measured transmission spectra for the micro-ring sensor array over time. The original transmission spectrum (a). Transmission spectra measured after 1 hour (b), 2 hours (c), 3 hours (d), 4 hours (e), 5 hours (f), and 6 hours (g).

In addition, the temperature change of water could also resonant frequency shifts. To examine this issue, we varied the temperature of the aqueous environment within a range from 25 - 40 °C by using a heating pad. Notably, it was observed that all 15 sensors in the array exhibited roughly the same frequency drifts along the same direction. This observation indicates that the variation in the temperature of the water does not scramble the spectrum of the sensor array. The mean resonant frequency shifts for these 15 resonant frequencies are plotted in Supplementary Fig. 13 as a function of temperature using blue dots, while their standard deviations are represented using red circles. As we can see from the figure, the frequency drifts increase linearly with the increased temperature. Through a linear fitting, the measured data results in a thermo-optic coefficient of about 25.9 pm/°C

(blue dashed line), which is in agreement with the one reported in the literature. Moreover, we found that the temperature variation in the room does not considerably change the spectrum of the sensor array.

Supplementary Fig. 13. The measured resonant wavelength shift for micro-ring sensors as a function of temperature. The blue dots and red circles represent the mean values and standard deviations from the measurements of these 15 micro-rings. The blue dashed line denotes the linear fitting of the mean.

4. *More information should be given on the coherence detection scheme to explain how the raw signal from a single coherent detector is transformed into an array. This should include showing the signals at each step of the process.*

Response: We thank the reviewer for making this valuable suggestion. In the revised main text, we added the section “**Code availability**” which includes detailed codes and example data to show the working flow of the DOFC-enabled interrogation method.

Code availability

Code of DOFC-enabled interrogation means for retrieving PA signals from the micro-ring sensor array can be accessed in public repository Zenodo [52].

Moreover, to better describe the coherent detection scheme, we also added the following paragraph to explain how the signal from a single coherent detector is transformed into the photoacoustic signals measured by the sensor array, in Supplementary Note 7 of the revised supplementary Information:

Notably, the transmission spectrum of the micro-ring sensor array, which carries the information of the ultrasonic wave, can be obtained by dividing $\tilde{D}_{\text{output}}(f)$ with respect to $D_{\text{input}}(f)$ as follows

$$T(f) = |\tilde{D}_{\text{output}}(f)/D_{\text{input}}(f)|^2 \tag{S7}$$

Then, we swept the transmission spectrum to locate the positions of the resonant frequencies f_i where i is the labeling for micro-ring sensors. Taking the sensor array with 15 elements as an example, $i = 1, 2, 3, \dots, 15$. In the absence of the ultrasonic wave, we recorded $f_i(0)$ as the original resonant frequency. When the ultrasonic wave interacts with the sensor array, all $f_i(t)$ start to change with time. In this condition, we define the time-dependent amplitude of the ultrasonic wave received by the i -th element $PA_i(t)$ as

$$PA_i(t) = f_i(t) - f_i(0) \quad (S8)$$

Thus, aided by the transmission spectrum measured through the DOFC, we can determine the time-dependent amplitude of the ultrasonic wave from all micro-ring sensors in parallel, without the need to lock single-wavelength lasers to the resonant frequencies as the conventional methods did.

5. *The authors claim that the PDMS layer was used only to protect the micro-ring and that the signal is a result of the high photoelastic coefficient of the glass. However, this contradicts the conclusion of Hazan et al. in silicon, where it was experimentally shown that the signal actually comes from the PDMS. The authors should substantiate their claim by using harder coating materials (e.g. normal silica, or PMMA) and show that the response does not change. If the coating determines the signal characteristics, it is possible that the authors were wrong about their conclusion and would need to modify the manuscript accordingly. In any case, the comparison to the alternative coating material is essential. If PDMS is essential for the performance obtained in this paper, the work of Hazan et al. should be properly cited.*

Response: We thank the reviewer for asking this important question. In the work of Hazan et al., the SiO₂-based micro-ring is much harder than the PDMS cladding, thus the signal comes from the PDMS. The situation is completely different in our work, where the micro-ring was made using soft chalcogenide material. To validate this conclusion by examining whether the PDMS cladding contributes to the measured signals, we performed additional experiments as suggested by the reviewer. In particular, we compared the performances of micro-ring sensors without cladding and with PDMS or SiO₂ claddings. The comparison shows that most of the signals are indeed contributed from chalcogenide-based micro-ring structures while the contribution from the elastic deformation in PDMS is small. To illustrate this point, in the revised submission, we added the following sentences in the section “**Structure of the sensor array**” of the revised main text: **A previous study showed that, in the architecture of silicon photonics, deformation in the PDMS cladding can contribute to the measured signals [19]. In Supplementary Note 5, we tested this effect with different types of cladding materials and found that the measured signals in this work are mainly contributed by chalcogenide-based micro-ring structures rather than PDMS claddings.**

In the revised supplementary Information, we also added a new section “**Supplementary Note 5**” to compare the performance of micro-ring sensors with different claddings and commented on the difference between our work and the work reported by Hazan et al.

Supplementary Note 5. Effects of claddings on chalcogenide-based micro-ring sensors

In this section, we describe the effects of claddings on the performance of chalcogenide-based micro-ring sensors. As a fair comparison, we chose three micro-ring sensors with roughly the same quality factors around 5×10^5 , and encapsulated them with different claddings. As the control group, one micro-ring sensor has no cladding, which is referred to as the null case. The other two sensors were encapsulated with 3- μm -thick claddings using either polydimethylsiloxane (PDMS) or silicon dioxide (SiO₂). Using the same optical system and the characterization procedure described in the

main text to produce Fig. 7(d), the amplitude maps of the measured signals as a function of time and translational distance are illustrated in Supplementary Fig. 4. As we can see from the figure, different claddings do not induce considerable differences in the amplitude maps and no strong surface acoustic wave is observed. Quantitatively, we found the peak values in these three cases are similar, indicating that signals are indeed contributed by the chalcogenide-based micro-ring structures. Therefore, we conclude that for micro-ring sensors made of soft chalcogenide-based material, the choices of cladding material do not considerably affect the performance of the sensor. It is worth noting that this observation is different from the one reported in Ref. [3] in which the sensor structure was made of a hard material SiO_2 . In that case, the deformation in relatively soft PDMS cladding contributes to the measured signal.

Supplementary Fig. 4. the amplitude maps of the measured signals as a function of time and translational distance. (a) With PDMS cladding. (b) With SiO_2 cladding. (c) Without cladding.

6. *The authors claim a 30-degree acceptance angle, referring to Fig. 4e. It would be good to elaborate on that. Could the authors show that time- and frequency-domain responses obtained at several angles (e.g. 10, 20, etc.). It would be good to see what happens beyond 30 degrees. The experimental results should be justified with a numerical simulation. The theoretical angular response of a microring is can be found in “Dong, B., Sun, C. & Zhang, H. F. Optical detection of ultrasound in photoacoustic imaging. *IEEE Trans. Biomed. Eng.* 64, 4–15 (2017).”*

Response: We thank the reviewer for making this valuable suggestion. We performed the characterization experiments again for the micro-ring sensor and show both the time and frequency responses at several angles of 10° , 20° , 30° , and 40° . A numerical investigation based on the theoretical framework described in the suggested reference was also provided. All new results were supplemented in the revised submission. In particular, in the section “**Characterization of the micro-ring sensor**” of the revised main text, we updated Fig. 7(e) and its corresponding caption as follows.

(e) Frequency response as a function of the acceptance angle. Two experimentally achieved -3 dB lines are also provided in white, matching the theoretical estimations in red.

We also revised the following description in the main context as: **By transforming time and position into frequency and acceptance angle, the frequency response of the micro-ring sensor as a function of the acceptance angle is shown in Fig. 7(e) (one-dimensional profiles of detailed time and frequency responses are provided in Supplementary Note 9). Moreover, experimentally achieved -3-dB lines are labeled in white, showing a similar structure and trend to the theoretically estimated ones in red (detailed in Supplementary Note 9 using the procedure described in Ref. [47]). These results indicate that for frequencies up to around 25 MHz, wide acceptance angles covering about ± 30 degrees can be achieved.**

In the revised supplementary Information, we also added a new section “Supplementary Note 9” to describe detailed procedures for numerical estimations on the acceptance angle. Detailed time and frequency responses at several angles of 10° , 20° , 30° , and 40° were also provided.

Supplementary Note 9. Acceptance angles for the micro-ring sensor

Optical ultrasound sensors generally support wide acceptance angles. In this section, we detailed the characterization process for the acceptance angle of the micro-ring sensor. The experimental setup has been shown in Fig. 7(a) of the main text. In the beginning, the relative distance between the ultrasonic source and the micro-ring sensor was about 5 mm, and the time and frequency responses at 0° are shown in Figs. 7(b) and (c) of the main text. Then, the micro-ring sensor was scanned in the horizontal direction. To generate a sufficient number of data to produce Fig. 7(d), the scanning step size was set to 5 μm . The translational distance can be converted into the acceptance angle through a simple trigonometric operation. The time responses at 10° , 20° , 30° , and 40° are provided in Supplementary Figs. 7(a)-(d). All signals were normalized to the maximum

value of Supplementary Fig. 7(a). As we can see from the figure, a larger receiving angle leads to a smaller peak amplitude and a larger time delay. More importantly, signals get broadened for larger angles, indicating smaller central frequencies and narrower operating bandwidths. To illustrate this point, frequency responses at these corresponding angles are provided in Supplementary Figs. 7(e)-(h). Quantitatively, the 3-dB bandwidths are 52 MHz, 21 MHz, 17 MHz, and 14 MHz at 10° , 20° , 30° , and 40° , respectively. Given that the 3-dB bandwidth at 0° is 115 MHz, we notice that the decreasing rate for the bandwidth of the sensor decays is fast for small angles and becomes slow for large angles.

Supplementary Fig. 7. Characterization for the acceptance angle of the micro-ring sensor. (a)-(d) Time responses of the micro-ring sensor with an acceptance angle at 10° , 20° , 30° , and 40° , respectively. (e)-(h) Frequency responses of the micro-ring sensor with an acceptance angle at 10° , 20° , 30° , and 40° , respectively.

We also numerically investigated the theoretical angular response of the micro-ring sensor. In particular, we followed the procedure described in Ref. [9] to examine the spatial distribution of ultrasonic detection. To comply with the parameters used in experiments, the diameter of the micro-ring sensor was set to $40 \mu\text{m}$ and the relative distance between the ultrasonic point source and the sensor at the beginning was 5 mm during simulations. The frequency response of the micro-ring sensor as a function of the acceptance angle is shown in Supplementary Fig. 8. Two red -3 dB lines are also provided for visualization purposes. This figure confirms the experimental results presented in Fig. 7(e) show similar trends to the theoretical ones.

Supplementary Fig. 8. Numerical investigation of the theoretical angular response of the micro-ring sensor.

7. *The authors should generally explain why their resolution is not higher (Fig. 4). Generally speaking, a 40-micron microring should give a 40-micron lateral resolution. Why is that not the case here? Also the axial resolution is lower than what I would expect for a 175 MHz bandwidth.*

Response: We thank the reviewer for asking this important question about imaging resolutions. Since the micro-ring sensor was placed at a distance from the carbon fiber and the sensor was also linearly translated with a very fine step size of $0.7\ \mu\text{m}$, the $40\text{-}\mu\text{m}$ diameter of the micro-ring sensor is not directly related to the lateral resolution of the imaging system. Instead, the lateral resolution should depend on the central frequency of the received acoustic signals. As for the axial resolution, the reviewer is correct that our measured value is lower than what can be achieved for this bandwidth. To address this issue, we performed experiments again to characterize the imaging resolutions with a much finer step size to avoid the effect of pixelation. In the revised main text, we updated these values in the section “**Characterization of the micro-ring sensor**” and discuss the discrepancy between the measured values and the theoretically estimated ones: **Theoretically, for partial view PAT, the lateral resolution is given by $0.71 v/(NAf_0) \approx 35.5\ \mu\text{m}$ [49]. Here, $v = 1,500\ \text{mm/ms}$ is the speed of sound, $NA \approx \sin(30^\circ) = 0.5$ is the numerical aperture estimated using the acceptance angle of the micro-ring sensor, and $f_0 = 60\ \text{MHz}$ is the central frequency. The small discrepancy between the estimated value ($35.5\ \mu\text{m}$) and the measured one ($50.4\ \mu\text{m}$) may be due to the inaccurate estimation of the numerical aperture. Moreover, the theoretically estimated axial resolution is given by $0.88v/\Delta f \approx 11.5\ \mu\text{m}$, where $\Delta f = 115\ \text{MHz}$ is the 3-dB bandwidth [49]. This value ($11.5\ \mu\text{m}$) is also smaller than the measured one ($18.9\ \mu\text{m}$), which is likely due to high-frequency components attenuating much more than low-frequency counterparts in agar that covers the carbon fiber.**

8. *The PSF shown in Fig. 4h is not so clear and does not look like the theoretical prediction (<https://doi.org/10.1364/AO.47.000561> -Fig. 9), which has also been verified by Hazan et al.*

Response: We thank the reviewer for this careful check and for providing this valuable reference. The point spread functions provided in the original submission suffer from a strong effect of pixelation, which is due to insufficient sampling during the scanning process. To address this issue and quantify the point spread function more accurately, we experimented again by using a much finer scanning step size ($0.7\ \mu\text{m}$). The newly obtained point spread function now looks similar to the one illustrated in the references the reviewer provided. In the revised main text, we modified the descriptions on the characterization of the point spread function and updated corresponding figures and captions in the section “**Characterization of the micro-ring sensor**”, as below: **This parameter was quantified by replacing the golden thin film in Fig. 7(a) with a horizontally placed carbon fiber ($6\ \mu\text{m}$ in diameter) buried inside the agar. The cross-section of the fiber was imaged to provide information on the point spread function of the imaging system. The micro-ring sensor was linearly translated with a step size of $0.7\ \mu\text{m}$ with a 2.8-mm range. UBP [40] was employed to synthesize detected PA signals at each position and produce the image in Fig. 7(h). One-dimensional profiles for quantifying both the lateral and axial resolutions are illustrated in Figs. 7(i) and (j), exhibiting full width at half maximum of about $50.4\ \mu\text{m}$ and $18.9\ \mu\text{m}$, respectively. These curves have similar shapes to the ones reported in the literature [19,34,48].**

...(h) Reconstructed images of the cross-section of the carbon fiber buried inside the agar. (i) One-dimensional profile along the lateral direction, suggesting a lateral resolution of $50.4\ \mu\text{m}$. (j) One-dimensional profile along the axial direction, suggesting an axial resolution of $18.9\ \mu\text{m}$.

9. *Using a rotation-based imaging system defies the logic of using optical sensors. As has been shown the UCL group, optical sensors have the advantage of working in a planar geometry. The authors need to show that their technique can do the same and still get good images, even if some*

resolution is lost.

Response: We thank the reviewer for his comments on performing imaging experiments in the planar geometry. Per the reviewer’s suggestion, instead of rotating the sample, we performed additional imaging experiments in the planar geometry. In particular, we linearly scanned the sensor array and successfully demonstrated imaging three interleaved black hairs. In the section “**Experimental result on imaging biological samples**” of the revised main text, we added the following paragraph to illustrate the imaging results obtained in the planar geometry:

Optical Sensors have the advantage of working in the planar geometry [41]. Here, instead of rotating samples, we show that our technique can function in the planar geometry as well. As shown in Fig. 8(a) (camera-captured image), three interleaved black hairs are chosen as the imaging target and are buried inside tissue-mimicking phantoms. Since 15 elements are not enough to suppress reconstruction artifacts in PAT, we linearly scanned the sensor array within a ± 8 -mm range and with a 20- μ m step size. The reconstructed image is shown in Fig. 8(b), showing excellent agreement with the original layout displayed in Fig. 8(a). Moreover, Figs. 8(c), (d), (e), and (f) show the reconstructed images using the information collected only within the range of (-8, -4) mm, (-4, 0) mm, (0, 4) mm, and (4, 8) mm, respectively. These results demonstrate that in contrast to the rotational geometry, different sensors in the planar geometry contain information through different projections.

Fig. 8. Experimental results of imaging interleaved black hairs using the micro-ring sensor array with parallel interrogation in the planar geometry. Scale bars: 1 mm. (a) Camera-captured image of the layout of three interleaved black hairs. (b) Reconstructed image by linearly scanning the sensor within a ± 8 -mm range and with a 20- μ m step size. (c)-(f) Reconstructed images using the information collected within the range of (-8, -4) mm, (-4, 0) mm, (0, 4) mm, and (4, 8) mm, respectively.

10. In Figs. 7a-d, it is not clear what the detection geometry was. Where the detector in-plane or arranged vertically. It feels like the benefit of using more detectors was not just from SNR, but also from having more projections. I suspect that the 1-sensor measurement would still not be as good as the one from the array even if the authors averaged. A fair comparison with the same number of signals is needed to answer this question.

Response: We thank the reviewer for asking questions regarding the detection geometry and different projections for elements in the sensor array. The first question the reviewer asks is about

the detection geometry of the leaf vein. In this work, the detection geometry for the leaf vein and the zebrafish is the same, in which the target biological sample and the sensor array lie in the same plane. To clarify this point, in the revised main text, we added the following sentence in the section “**Experimental result on imaging biological samples**”: **Both the biological sample and the micro-ring sensor array lie in the same horizontal plane, which is detailed and sketched in Supplementary Note 1.**

In the revised Supplementary Information, we modified the section “Supplementary Note 1” to illustrate detailed imaging geometry for biological samples.

Supplementary Note 1. Detailed imaging procedures for biological samples

The experimental setup for imaging biological tissue is schematically shown in Supplementary Fig. 1. The light source was chosen as a 532-nm laser (Beamtech, Dawa 100) with a pulse width of 6.5 ns and a repetition rate of 10 Hz. An optical diffuser (Thorlabs, DG10-120) was used to expand and homogenize illuminating light. The light illuminated the sample from the top with an area of about 8 mm in diameter. To mitigate artifacts due to the limited view, the water tank was mounted on a motorized rotational stage, which rotated with a step size of 1 degree. The sensor array was hung in water at the same horizontal plane and was about 4 cm away from the edge of the sample. In this condition, photoacoustic tomography (PAT) was performed by rotating the zebrafish while keeping the sensor array and excitation light fixed. When the pulsed light irradiated the sample, each micro-ring sensor produced an A-line signal, which contains 4,096 data points. After rotating 360 degrees, universal back projection [1] was employed to reconstruct the image through these 360 sets of A-line signals. To acquire one image, the scanning process took about 36 seconds (360 optical pulses). A microcontroller (STMicroelectronics, STM32) was employed to synchronize the motorized rotational stage, the laser, and the data acquisition process.

Supplementary Fig. 1. Experimental setup of performing photoacoustic tomography for biological samples. M: mirror; PH: pinhole; L: lens; SA: sensor array; WT: water tank; DOFC: digital optical frequency comb; EDFA: erbium-doped fiber amplifier; CR: coherent receiver; DSP: digital signal processing.

The second question the reviewer asks is about the difference in projections for different micro-ring sensors. In our experimental setup, the biological samples, including both the leaf and zebrafish, are

in centimeter scales. Since the center-to-center distance between different micro-ring sensors is only 400 μm and the sample was rotated during experiments, all micro-ring sensors, regardless of the ones at the edge or at the center, share similar views to the sample. Thus, we could hardly discern any considerable difference in the images reconstructed by different single micro-ring sensors. To clarify this point, in the revised main text, we added the following sentence in the section “**Experimental result on imaging biological samples**”: *As a comparison, the reconstructed image using a typical micro-ring sensor is shown in Fig. 3(b). More reconstructed images using other single micro-ring sensors are provided and compared in Supplementary Note 3, exhibiting similar performance in such a rotation-based detection geometry.*

In the revised supplementary Information, we also added a new section “Supplementary Note 3” to illustrate reconstructed images of the leaf vein using different 1-sensor measurements.

Supplementary Note 3. Imaging reconstruction of the leaf vein with 1-sensor measurement

The reconstructed image in Fig. 3(b) of the main text was obtained using the 8th sensor in the micro-ring sensor array, which is the one at the center. Normally, different elements in the sensor array maintain different views of the sample. In this condition, a coherent summation of the information captured through these elements can improve the image quality by reducing artifacts due to a limited view. Nonetheless, since the center-to-center distance between different micro-ring sensors is only 400 μm and the sample was rotated during experiments, all micro-ring sensors, regardless of the ones at the edge or at the center, share roughly the same views of the sample. This fact can be validated by examining the reconstructed images of the leaf vein achieved through different 1-sensor measurements. In particular, Supplementary Figs. 3(a), (b), (c), and (d) show the reconstructed images using only the 1st, the 4th, the 12th, and the 15th element, respectively. As shown in the figure, we could hardly see any considerable difference in the images reconstructed by different single micro-ring sensors. Compared with the one shown in the main text, the fluctuations in terms of the contrast-to-noise ratio of these images were within 7%, which we believe is most likely due to the variation in the sensitivity of different sensing elements.

Supplementary Fig. 3. Reconstructed images of the leaf vein through different elements in the sensor array. Scale bars: 1 mm. (a) The 1st element; (b) the 4th element; (c) the 12th element; (d) the 15th element.

Although different sensor elements in the rotation geometry share roughly the same view of the sample, the condition is completely different when performing imaging in the planar geometry.

In other words, using more sensor elements in the planar geometry generally benefits from having different projections, as pointed out by the reviewer. To illustrate this point, in the section “**Experimental result on imaging biological samples**” of the revised main text, we added the following paragraph to show the difference in projections of different sensor elements:

Optical Sensors have the advantage of working in the planar geometry [41]. Here, instead of rotating samples, we show that our technique can function in the planar geometry as well. As shown in Fig. 8(a) (camera-captured image), three interleaved black hairs are chosen as the imaging target and are buried inside tissue-mimicking phantoms. Since 15 elements are not enough to suppress reconstruction artifacts in PAT, we linearly scanned the sensor array within a ± 8 -mm range and with a 20- μ m step size. The reconstructed image is shown in Fig. 8(b), showing excellent agreement with the original layout displayed in Fig. 8(a). Moreover, Figs. 8(c), (d), (e), and (f) show the reconstructed images using the information collected only within the range of (-8, -4) mm, (-4, 0) mm, (0, 4) mm, and (4, 8) mm, respectively. These results demonstrate that in contrast to the rotational geometry, different sensors in the planar geometry contain information through different projections.

Fig. 8. Experimental results of imaging interleaved black hairs using the micro-ring sensor array with parallel interrogation in the planar geometry. Scale bars: 1 mm. (a) Camera-captured image of the layout of three interleaved black hairs. (b) Reconstructed image by linearly scanning the sensor within a ± 8 -mm range and with a 20- μ m step size. (c)-(f) Reconstructed images using the information collected within the range of (-8, -4) mm, (-4, 0) mm, (0, 4) mm, and (4, 8) mm, respectively.

11. *It is not clear what the imaging geometry of the zebrafish was.*

Response: We thank the reviewer for asking this question and allowing us to clarify this issue. The imaging geometry of the zebrafish is the same as that of the leaf vein. To clarify this point, in the revised main text, we added the following sentence in the section “**Experimental result on imaging biological samples**”: **Both the biological sample and the micro-ring sensor array lie in the same horizontal plane, which is detailed and sketched in Supplementary Note 1.**

In the revised Supplementary Information, we modified the section “Supplementary Note 1” to illustrate detailed imaging geometry for biological samples.

Supplementary Note 1. Detailed imaging procedures for biological samples

The experimental setup for imaging biological tissue is schematically shown in Supplementary Fig. 1. The light source was chosen as a 532-nm laser (Beamtech, Dawa 100) with a pulse width of 6.5 ns and a repetition rate of 10 Hz. An optical diffuser (Thorlabs, DG10-120) was used to expand and homogenize illuminating light. The light illuminated the sample from the top with an area of about 8 mm in diameter. To mitigate artifacts due to the limited view, the water tank was mounted on a motorized rotational stage, which rotated with a step size of 1 degree. The sensor array was hung in water at the same horizontal plane and was about 4 cm away from the edge of the sample. In this condition, photoacoustic tomography (PAT) was performed by rotating the zebrafish while keeping the sensor array and excitation light fixed. When the pulsed light irradiated the sample, each micro-ring sensor produced an A-line signal, which contains 4,096 data points. After rotating 360 degrees, universal back projection [1] was employed to reconstruct the image through these 360 sets of A-line signals. To acquire one image, the scanning process took about 36 seconds (360 optical pulses). A microcontroller (STMicroelectronics, STM32) was employed to synchronize the motorized rotational stage, the laser, and the data acquisition process.

Supplementary Fig. 1. Experimental setup of performing photoacoustic tomography for biological samples. M: mirror; PH: pinhole; L: lens; SA: sensor array; WT: water tank; DOFC: digital optical frequency comb; EDFA: erbium-doped fiber amplifier; CR: coherent receiver; DSP: digital signal processing.

12. *What was the fiber-to-fiber insertion loss of the system? Could the authors elaborate on their fiber-to-chip bonding procedure?*

Response: We thank the reviewer for asking this technical question. We supplemented the information regarding the fiber-to-chip insertion loss of the system in the section “**Structure of the sensor array**” of the revised main text: **Currently, the fiber-to-waveguide insertion loss was estimated at around 6 dB for each side, indicating a total 12-dB insertion loss for the entire sensor**

chip. Detailed procedures regarding fiber-to-chip bonding can be found in Supplementary Note 6.

In the revised Supplementary Information, we also added a new section “Supplementary Note 6” to describe detailed procedures for fiber-to-chip bonding.

Supplementary Note 6. The procedures for fiber-to-chip bonding

In this section, we describe the procedures for fiber-to-chip bonding. The sensor chip was placed at a coupling platform, which was monitored by a microscope. The magnification of the eyepiece and objective lenses of the microscope were chosen to be 12× and 20×, respectively. Based on the shape and size of the propagating mode inside the bus waveguide, we chose to use a single-mode fiber with a mode fiber diameter of $3.2 \pm 0.3 \mu\text{m}$ at 1550 nm (Nufern UHNA7, Coherent). Aided by the microscope, both ends of the bus waveguide were roughly aligned to the aforementioned type of single-mode fibers, as shown in Supplementary Fig. 5. This procedure was accomplished by adjusting the three-axis high-precision translation stage (MAX311D/M, Thorlabs) underneath the fiber. Note that a small gap was left between the waveguide and fiber for fine-tuning afterward.

Supplementary Fig. 5. A schematic illustration of the fiber-to-chip bonding structure.

Then, the input side of the fiber was connected to a continuous-wave laser (Keysight, 8164B, 10-kHz linewidth, 1,550 nm), while the output side of the fiber was attached to an optical power meter (PM100D, Thorlabs) for fine-tuning. The two three-axis stages that support the two fibers were adjusted consecutively. When gradually decreasing the gap between the fiber and the waveguide along the z direction, the positions along the x and y directions were slightly adjusted to maximize the measurement of the power meter. As the fiber and the waveguide fit tightly, a small amount of ultraviolet curing adhesive (NOA61, Norland) with a refractive index of 1.56 was dripped at the connection point. An ultraviolet curing lamp (NVSUA U365nm, Nichia) with an illumination spot of 1.5 mm and a wavelength of 365 nm was used for solidifying the curing adhesive. Empirically, we chose an illumination time of 5 minutes and an illumination distance of 5 cm to guarantee satisfactory performance. After this fine-tuning procedure, we experimentally quantified the insertion loss between the fiber and the waveguide was about 6 dB, leading to a total insertion loss of 12 dB for the sensor chip.

13. *Could the authors provide more information on how the black leaf was produced? Namely, what was the ink, was it diluted, and what was the infusion protocol? A photo of the leaf would be beneficial.*

Response: We thank the reviewer for this valuable suggestion. We supplemented the information on how to produce the leaf in the revised submission. In particular, in the revised main text, we added the following sentence in the section “**Experimental result on imaging biological samples**”:

The preparation process of the leaf can be found in Supplementary Note 2.

In the revised Supplementary material, we also added a new section “Supplementary Note 2” to describe the protocol to prepare the leaf.

Supplementary Note 2. The protocol to prepare the leaf

This protocol describes the detailed procedure to prepare the leaf for imaging purposes. The main goal of these procedures is to remove mesophyll through chemical corrosion using acidic or alkaline substances. The leaf veins, on the other hand, are kept and stained for imaging purposes. The protocol is illustrated as follows:

1. Preparing solution: add 20 g sodium hydroxide and 10 g sodium carbonate into 500 ml water and stir them evenly.
2. Heating solution: when the solution is about to boil, add a piece of a diamond leaf. While keeping the solution slightly boiling, heat the diamond for 5 minutes.
3. Removing mesophyll: take out the diamond leaf and put it into clear water. Use a brush to gently remove the mesophyll along the directions of the veins. Then, clean the leaf with clean water and dry the water again.
4. Bleaching: put the leaf into sodium hypochlorite solution for bleaching, wash them with clean water after bleaching, and dry the water again.
5. Staining: dilute the ink 1:1 with water and put the leaf into the solution for staining. After 3-4 minutes, take the leaf out. Rinse the leaf with clean water and dry the water.

After performing the above five steps, the leaf is ready for imaging purposes. A photo of the leaf is shown in Supplementary Fig. 2.

Supplementary Fig. 2. A photo of the leaf used for photoacoustic imaging.

14. What was the fabrication procedure of the gold layer? What type of machine was used to deposit the gold layer and how long did it take? The authors could use the following paper as reference: Seeger, M., Soliman, D., Aguirre, J. et al. Pushing the boundaries of optoacoustic microscopy by total impulse response characterization. *Nat Commun* 11, 2910 (2020).

Response: We thank the reviewer for asking this question. We supplemented the information on the fabrication procedure of the gold layer in the revised submission. In particular, in the revised main text, we added the following sentence in the section “**Characterization of the micro-ring sensor**”: ... onto a 200-nm-thick golden thin film (fabrication procedure described in Supplementary Note 8) [27, 30, 46]

In the revised Supplementary Information, we also added a new section “Supplementary Note 8” to describe the fabrication procedure.

Supplementary Note 8. The fabrication procedure for the golden layer

The fabrication procedure for the golden layer used for photoacoustic characterization is similar to that reported in Ref. [8]. The golden layer was fabricated under a high vacuum below 5×10^{-9} Torr using electron beam assisted deposition (DE400DUL, Detech). The substrate was chosen as Silicon dioxide with a thickness of 1,500 μm . Then, a 5-nm layer of titanium with a purity $> 99.99\%$ ($\Phi 60 \times 2$ mm, ZhongNuo Advanced Material (Beijing) Technology Co., Ltd) was deposited at 0.3 $\text{\AA}/\text{s}$ on the substrate, serving as the adhesion layer. Subsequently, a 200-nm layer of gold with a purity $> 99.99\%$ ($\Phi 60 \times 2$ mm, ZhongNuo Advanced Material (Beijing) Technology Co., Ltd) was deposited at the same speed. Such a relatively low deposition speed is to guarantee the high crystalline quality of the deposited metal layers. The entire procedure took about 2 hours.

15. *What was approximately the diameter of the focused optical beam used to generate the optoacoustic point source?*

Response: We thank the reviewer for asking this important question. In the revised main text, we supplemented the information in the section “**Characterization of the micro-ring sensor**”: **The diameter of the focal spot was about 9 μm .**

16. *What was the laser power, and how much was coupled into the waveguide?*

Response: We thank the reviewer for asking this important technical question. The laser out power was set to 10 mW and about 0.12 mW was coupled into the waveguide. In the revised main text, we supplemented this information in the section “**Experimental setup for PAT using the sensor array**”

- a. ... and its output power was set to 10 mW.
- b. The signal beam (~ 0.12 mW in the bus waveguide) interacted with the sensor array ...

Reviewer #2:

This work has made two major contributions to optical detection of photoacoustics. First the authors demonstrated 15-microring sensor array made of chalcogenide glass for PAT imaging. The sensors show high quality factor (10^5), high bandwidth (175 MHz) and low NEP ($2.2 \text{ mPa}\sqrt{\text{Hz}}$). Second, this work synthesizes digital optical frequency comb to interrogate the sensor array in parallel fashion with only one source and sensor. This method greatly simplifies the interrogation method and enables high speed photoacoustic tomography with microring array sensor. Photoacoustic imaging of fast-moving object, leaf veins and live zebra fish have been demonstrated with the proposed method.

1. Overall, this is an excellent work. A few questions are listed below for authors' clarification:

Response: We thank the reviewer for recognizing our work.

2. In line 122, what is the photoelastic coefficient of the material? Only young's modulus of 31.9 GPa is given.

Response: We thank the reviewer for asking this important question. We supplemented this information in the section “**Structure of the sensor array**” of the revised main text: This material has a good photoelastic property with Young's modulus of 31.9 GPa and photoelastic coefficient of about 0.238, leading to a good sensitivity to ultrasound.

3. The signal comes primarily from refractive index change of the microring (photoelastic effect), right? does the elastic deformation of the PDMS coating contribute to the detection?

Response: We thank the reviewer for asking this important question. Since the chalcogenide material used to make the micro-ring structure is soft, we found the signal comes primarily from the refractive index change of the micro-ring. To support this claim, we characterized the performance between micro-ring sensors with and without PDMS cladding and did not find a significant difference. Therefore, we conclude that most of the signals are indeed contributed from chalcogenide-based micro-ring structures while the contribution from the elastic deformation in PDMS is negligible. To illustrate this point, in the revised submission, we added the following sentences in the section “**Structure of the sensor array**”: A previous study showed that, in the architecture of silicon photonics, deformation in the PDMS cladding can contribute to the measured signals [19]. In Supplementary Note 5, we tested this effect with different types of cladding materials and found that the measured signals in this work are mainly contributed by chalcogenide-based micro-ring structures rather than PDMS claddings.

In the revised supplementary Information, we also added a new section “**Supplementary Note 5**” to compare the performance of micro-ring sensors with different claddings and commented on

the difference between our work and the work reported by Hazan et al.

Supplementary Note 5. Effects of claddings on chalcogenide-based micro-ring sensors

In this section, we describe the effects of claddings on the performance of chalcogenide-based micro-ring sensors. As a fair comparison, we chose three micro-ring sensors with roughly the same quality factors around 5×10^5 , and encapsulated them with different claddings. As the control group, one micro-ring sensor has no cladding, which is referred to as the null case. The other two sensors were encapsulated with 3- μm -thick claddings using either polydimethylsiloxane (PDMS) or silicon dioxide (SiO_2). Using the same optical system and the characterization procedure described in the main text to produce Fig. 7(d), the amplitude maps of the measured signals as a function of time and translational distance are illustrated in Supplementary Fig. 4. As we can see from the figure, different claddings do not induce considerable differences in the amplitude maps and no strong surface acoustic wave is observed. Quantitatively, we found the peak values in these three cases are similar, indicating that signals are indeed contributed by the chalcogenide-based micro-ring structures. Therefore, we conclude that for micro-ring sensors made of soft chalcogenide-based material, the choices of cladding material do not considerably affect the performance of the sensor. It is worth noting that this observation is different from the one reported in Ref. [3] in which the sensor structure was made of a hard material SiO_2 . In that case, the deformation in relatively soft PDMS cladding contributes to the measured signal.

Supplementary Fig. 4. the amplitude maps of the measured signals as a function of time and translational distance. (a) With PDMS cladding. (b) With SiO_2 cladding. (c) Without cladding.

4. In figure 2(b), is the one in the center the bus waveguide and are the 2 on the edges the microring? Why does the cross-section profile look like this? It is better to indicate the region on figure 2(a).

Response: We thank the reviewer for asking this important question. In the original Fig. 2(b), the center one is the tapered end of the bus waveguide while the other two on the edges are the unexposed film. Therefore, the way we present the bus waveguide can be quite misleading. To avoid confusion, we modified this figure in the revised main text, which shows only the cross-section of the bus waveguide in the image. We also modified the colormap of Fig. 2(a) to make it consistent with the one in Fig. 2(b).

5. *DOFC is described mathematically in the SI. It will be helpful to readers if the authors can provide a short intuitive understanding of its working principle in the main text. This is one of the most important contributions of the work.*

Response: We thank the reviewer for this valuable suggestion. To provide an intuitive understanding of the working principle of the DOFC, we added the following sentences in the section “**Experimental setup for PAT using the sensor array**” of the main text: **Pictorially, the DOFC exhibits a comb structure with a fine comb tooth in the frequency domain, which can be measured through a single coherent detector in the time domain. By quantifying the changes in amplitude and phase of these comb teeth, the transmission spectrum of the sensor array in the frequency domain can be accurately determined. By scrutinizing the temporal change in the transmission spectrum, time-dependent acoustic signals can be determined.**

6. *In the section of “Characterization of the micro-ring sensor”, it mentions the NEP is 7.1 Pa within 20-MHz. Is it an averaged value of the 15 sensors of the array? What is the uniformity of the sensitivity and the NEP of the 15 sensors.*

Response: We thank the reviewer for raising this valuable question regarding the uniformity of the sensors in the array. Due to the finite fabrication precision, these micro-ring sensors hold slightly different quality factors, leading to slightly different performances. The mentioned NEP is measured from a typical micro-ring sensor with a quality factor ranked in the middle. The sensitivities and NEPs for other sensors are within $\pm 10\%$ of the presented one. To clarify this issue, in the revised main text, we supplemented this information in the section “**Characterization of the micro-ring sensor**” as follows:

a. **For fair assessment, we describe the characterization process for the micro-ring sensor with the**

8th largest quality factor in detail. For this typical micro-ring sensor, ...

- b. All other micro-ring sensors were characterized in the same way, exhibiting NEPs within $\pm 10\%$ of the value reported above.

7. *It mentions that the NEP of the sensor is comparable to the state-of-the-art ultrasound sensor. How about the parallel interrogation method compared with conventional method that is used in reference [10]? Does the parallel interrogation method introduce extra noise and change the NEP?*

Response: We thank the reviewer for asking this important question. The parallel interrogation method does introduce extra noise to the sensor and changes the NEP. Experimentally, for the same micro-ring sensor that produces Fig. 7, the parallel interrogation method increases the NEP to about 36.1 Pa within a 20-MHz bandwidth. The NEPs for the rest of the micro-ring sensors are within the $\pm 10\%$ range. These values are comparable with the NEP of the sensor array reported in Ref. [10], which was 24 Pa within a 10-MHz bandwidth. In the revised main text, we supplemented this information in the section “**Characterization of the micro-ring sensor.**”: *It should be noted that the use of the parallel interrogation method introduces extra noise and thus increases the NEP. For the same micro-ring sensor, the NEP increases to about 36.9 Pa within a 20-MHz bandwidth (Detailed in Supplementary Note 11). The NEPs for the rest of the micro-ring sensors are within the $\pm 10\%$ range. Nonetheless, these values are still comparable to the NEP reported with conventional methods in Ref. [10] (24 Pa within a 10-MHz bandwidth).*

In the revised Supplementary Information, we also added a new section “**Supplementary Note 11**” to show the detailed quantification process of the NEP with parallel interrogation.

Supplementary Note 11. Noise-equivalent pressure for the parallel interrogation method

The parallel interrogation method certainly introduces additional noise, thus increasing the NEP of the measurement. In this section, we detail the quantification process of the NEP with the parallel interrogation method. Following the similar procedure described in Supplementary Note 10, the amplitude spectral density of the micro-ring sensor $M^{\text{PI}}(f)$ is shown in Supplementary Fig. 10(a). It is worth mentioning that the parallel interrogation method adopts the spectral shift (in the unit of MHz) as the indicator for the strength of ultrasound. By using the same calibrated needle hydrophone, the sensitivity of the micro-ring sensor $S_M^{\text{PI}}(f)$ was estimated and shown in Supplementary Fig. 10(b). Similarly, the noise amplitude spectral density $N_V^{\text{PI}}(f)$ was also quantified using the spectral shift, which is shown in Supplementary Fig. 10(c). With these parameters, the NEP spectral density $N_{\text{Pa}}^{\text{PI}}(f)$ can be estimated as

$$N_{\text{Pa}}^{\text{PI}}(f) = N_V^{\text{PI}}(f) / S_M^{\text{PI}}(f) \quad (\text{S13})$$

This parameter is shown in Supplementary Fig. 10(d). The RMS pressure within the range from 0 to 20 MHz was computed to be 36.9 Pa, which is larger than the NEP of a single micro-ring quantified above using the conventional approach.

Supplementary Fig. 10. Quantification of the noise-equivalent pressure (NEP) of the micro-ring sensor with parallel interrogation. (a) The amplitude spectral density $M^{PI}(f)$. (b) The sensitivity of the micro-ring sensor $S_M^{PI}(f)$. (c) The noise amplitude spectral density $N_V^{PI}(f)$. (d) The NEP spectral density $N_{Pa}^{PI}(f)$.

8. *Since the parallel interrogation is a relatively new method. It will be helpful to include some benefits/ challenges of this method.*

Response: We thank the reviewer for this valuable suggestion. We added the following sentences in the section “**Discussion**” of the revised main text to describe the benefits brought by the parallel interrogation method: **We also developed an effective means of parallel interrogation of the micro-ring sensor array using only one source-detector pair. In contrast to previously demonstrated micro-ring sensor arrays that typically require one source-detector pair per channel [27], the developed means of parallel interrogation can greatly simplify the system setup while speeding up the data acquisition process. Such a simplification is particularly valuable for developing head-mount imaging devices with the optical ultrasound sensor array, where both compact size and fast data acquisition are required. Moreover, the means of parallel interrogation eliminates the necessity to synchronize the signals measured by each element, which benefits the processes of data collection and image reconstruction for PAT. With these advantages, ...**

We also added the following sentences in the section “**Discussion**” of the revised main text to describe the challenges the parallel interrogation method encountered: **Besides the perspective of design, the practical challenge of parallel interrogation also comes from the energy loss of the sensor array. To employ the same interrogation means for the two-dimensional sensor array, a bus waveguide needs to be designed with a relatively long zigzag path to couple all micro-rings distributed in the two-dimensional plane. Given inevitable coupling loss and a 0.2-dB/cm energy loss of the bus waveguide based on the current fabrication technique, these micro-ring sensors will**

operate in considerably different conditions. The demodulation speed of the DOFC is another important issue that needs to be considered for real-time imaging. Currently, it took us 0.05 seconds to demodulate an acoustic signal with a length of 10 μ s (large enough to cover the field of view) for the sensor array with 15 elements (Intel(R) Core(TM) i7-10700 CPU @ 2.90GHz and 32 GB RAM), which can catch up with the laser repetition rate for real-time photoacoustic tomography (10 Hz). To incorporate more elements, we may need to optimize the demodulation codes and upgrade the computational facility.

Reviewer #3:

In this manuscript, Pan et al. report the fabrication of optical ultrasound sensors using chalcogenide compound materials. In particular, a sensor array that contains 15 microrings was demonstrated for photoacoustic tomography. The parameters of these microring sensors, such as sensitivity and bandwidth, are attractive to the community of photoacoustic imaging. Moreover, to enable parallel interrogation by using all these elements simultaneously, the authors further introduced a new and novel means by developing a digital optical frequency comb with an ultrafine comb tooth. With parallel interrogation, photoacoustic imaging for fast moving objects and live animals was demonstrated, confirming the validity of this work. In general, this is thorough and valid work. Given the means of parallel interrogation of optical ultrasound sensor array was rarely reported and in urgent need for the field of optical ultrasound sensor enabled photoacoustic imaging, this paper represents an important step forward. The paper is well written and fairly complete in terms of data. Still, I have a few questions

1. *The sensitivity of the micro-ring sensor was quantified to have a very small NEP down to 7.1 Pa. Although this value seems to be attractive, however, it seems to be the property of the optical ultrasound sensors only. When employing the DOFC-enabled parallel interrogation, will additional noise occur? If yes, please provide this situation in more detail, as it may become one of the limitations of this interrogation means. Generally speaking, using a more complex measurement scheme usually introduces more noise.*

Response: We thank the reviewer for asking this important question. The parallel interrogation method does introduce extra noise to the sensor and changes the NEP. Experimentally, for the same micro-ring sensor that produces Fig. 7, the parallel interrogation method increases the NEP to about 36.9 Pa within a 20-MHz bandwidth. The NEPs for the rest of the micro-ring sensors are within the $\pm 10\%$ range. These values are comparable with the NEP of the sensor array reported in Ref. [10], which was 24 Pa within a 10-MHz bandwidth. These values are comparable with the NEP of the sensor array reported in Ref. [10], which was 24 Pa within a 10-MHz bandwidth. In the revised main text, we supplemented this information in the section “**Characterization of the micro-ring sensor.**”: *It should be noted that the use of the parallel interrogation method introduces extra noise and thus increases the NEP. For the same micro-ring sensor, the NEP increases to about 36.9 Pa within a 20-MHz bandwidth (Detailed in Supplementary Note 11). The NEPs for the rest of the micro-ring sensors are within the $\pm 10\%$ range. Nonetheless, these values are still comparable to the NEP reported with conventional methods in Ref. [10] (24 Pa within a 10-MHz bandwidth).*

In the revised Supplementary Information, we also added a new section “**Supplementary Note 11**” to show the detailed quantification process of the NEP with parallel interrogation.

Supplementary Note 11. Noise-equivalent pressure for the parallel interrogation method

The parallel interrogation method certainly introduces additional noise, thus increasing the NEP of the measurement. In this section, we detail the quantification process of the NEP with the

parallel interrogation method. Following the similar procedure described in Supplementary Note 10, the amplitude spectral density of the micro-ring sensor $M^{\text{PI}}(f)$ is shown in Supplementary Fig. 10(a). It is worth mentioning that the parallel interrogation method adopts the spectral shift (in the unit of MHz) as the indicator for the strength of ultrasound. By using the same calibrated needle hydrophone, the sensitivity of the micro-ring sensor $S_M^{\text{PI}}(f)$ was estimated and shown in Supplementary Fig. 10(b). Similarly, the noise amplitude spectral density $N_V^{\text{PI}}(f)$ was also quantified using the spectral shift, which is shown in Supplementary Fig. 10(c). With these parameters, the NEP spectral density $N_{Pa}^{\text{PI}}(f)$ can be estimated as

$$N_{Pa}^{\text{PI}}(f) = N_V^{\text{PI}}(f)/S_M^{\text{PI}}(f) \quad (\text{S13})$$

This parameter is shown in Supplementary Fig. 10(d). The RMS pressure within the range from 0 to 20 MHz was computed be to 36.9 Pa, which is larger than the NEP of a single micro-ring quantified above using the conventional approach.

Supplementary Fig. 10. Quantification of the noise-equivalent pressure (NEP) of the micro-ring sensor with parallel interrogation. (a) The amplitude spectral density $M^{\text{PI}}(f)$. (b) The sensitivity of the micro-ring sensor $S_M^{\text{PI}}(f)$. (c) The noise amplitude spectral density $N_V^{\text{PI}}(f)$. (d) The NEP spectral density $N_{Pa}^{\text{PI}}(f)$.

2. When quantifying the resolutions in Fig. 4, the image seems to be quite pixelated. The authors should spend more effort on determining these values more accurately. Also, the vertical axis of the -6-dB bandwidth seems to be mistakenly labeled. Please correct.

Response: We thank the reviewer for this careful check. The point spread functions provided in the original submission suffer from a strong effect of pixelation, which is due to insufficient sampling during the scanning process. To address this issue and quantify the point spread functions more accurately, we performed the experiment again by using a much finer scanning step size (0.7 μm). In the revised main text, we modified the descriptions on the characterization of the point spread function and updated corresponding figures and captions in the section “**Characterization of the micro-ring sensor**”, as below: **This parameter was quantified by replacing the golden thin film in**

Fig. 7(a) with a horizontally placed carbon fiber (6 μm in diameter) buried inside the agar. The cross-section of the fiber was imaged to provide information on the point spread function of the imaging system. The micro-ring sensor was linearly translated with a step size of 0.7 μm with a 2.8-mm range. UBP [40] was employed to synthesize detected PA signals at each position and produce the image in Fig. 7(h). One-dimensional profiles for quantifying both the lateral and axial resolutions are illustrated in Figs. 7(i) and (j), exhibiting full width at half maximum of about 50.4 μm and 18.9 μm , respectively. These curves have similar shapes to the ones reported in the literature [19,34,48].

...(h) Reconstructed images of the cross-section of the carbon fiber buried inside the agar. (i) One-dimensional profile along the lateral direction, suggesting a lateral resolution of 50.4 μm . (j) One-dimensional profile along the axial direction, suggesting an axial resolution of 18.9 μm .

We also corrected the labeling of the vertical axis with respect to the -6-dB bandwidth in Fig. 7(c) of the revised main text:

3. The experimentally achieved axial and lateral resolutions are 24 and 80 μm , respectively. These values seem to be inconsistent with the central frequency (60 MHz) and bandwidth (175 MHz) of the microring sensor. The authors should either provide detailed explanations on justifying this

discrepancy or correct these values.

Response: We thank the reviewer for asking this important question. The reviewer is correct that our measured values are worse than what can be estimated from the central frequency and the bandwidth. To answer this question, we performed experiments again to characterize the imaging resolutions with a much finer step size to avoid the effect of pixelation and also discuss potential factors that may cause this discrepancy. In the revised main text, we updated these values in the section “**Characterization of the micro-ring sensor**” and discuss the discrepancy between the measured values and the theoretically estimated ones: **Theoretically, for partial view PAT, the lateral resolution is given by $0.71 v/(NAf_0) \approx 35.5 \mu\text{m}$ [49]. Here, $v = 1,500 \text{ mm/ms}$ is the speed of sound, $NA \approx \sin(30^\circ) = 0.5$ is the numerical aperture estimated using the acceptance angle of the micro-ring sensor, and $f_0 = 60 \text{ MHz}$ is the central frequency. The small discrepancy between the estimated value ($35.5 \mu\text{m}$) and the measured one ($50.4 \mu\text{m}$) may be due to the inaccurate estimation of the numerical aperture. Moreover, the theoretically estimated axial resolution is given by $0.88v/\Delta f \approx 11.5 \mu\text{m}$, where $\Delta f = 115 \text{ MHz}$ is the 3-dB bandwidth [49]. This value ($11.5 \mu\text{m}$) is also smaller than the measured one ($18.9 \mu\text{m}$), which is likely due to high-frequency components attenuating much more than low-frequency counterparts in agar that covers the carbon fiber.**

4. *The comb tooth of the digital optical frequency comb is 39 MHz. The authors provide little information on the choice of this value. Could we use a larger or smaller comb tooth? What are the possible affections on the imaging performance?*

Response: We thank the reviewer for raising this important question. The choice of the comb tooth was based on the quality factor of the micro-ring sensors. To better illustrate the criterion of choosing the comb tooth, we added the following descriptions in the section “**Discussion**” of the revised main text: **Moreover, it should be noted that the comb tooth in this study (39.0625 MHz) was chosen based on the quality factors of these micro-ring sensors. For quality factors within the range of $5 \sim 7 \times 10^5$, the full widths at half maximum of these resonant frequencies are typically $277 \sim 386 \text{ MHz}$, meaning that $7 \sim 10$ sampling points were adopted to locate the precise location of the resonant frequencies. In practice, for micro-ring sensors with small quality factors, a denser frequency comb could be used, which reduces the number of sampling points and alleviates the computational burden. In contrast, micro-ring sensors with high quality factors generally require a finer frequency comb to locate the positions of the resonant frequencies with high accuracy.**

5. *I suggest the authors estimate the moving velocity of fast-moving objects, which demonstrates the capability of the micro-ring sensor array and its means of parallel interrogation means.*

Response: We thank the reviewer for making this valuable suggestion. We supplemented this

information in the section “**Experimental results on imaging fast-moving objects in a single-shot measurement**” of the revised main text to demonstrate the capability of the micro-ring sensor array in imaging fast-moving objects: **By quantifying the traveling distance within a given time interval, the moving speed of the microsphere was estimated to be 4.8 mm/s, which is close to the speed of flowing water in the tube.**

6. *The resolution of live animals does not seem to be as good as tens of micrometers. This is natural as I believe high-frequency ultrasound may not propagate well in thick scattering samples or not even be fully excited in zebrafish. If this is the case, the authors should describe this issue. In other words, we may not need such a high-frequency bandwidth for imaging thick biological samples. A detailed discussion on this issue may be helpful to researchers in the field to design their optical ultrasound sensors in the future.*

Response: We thank the reviewer for raising this important question. We agree with the reviewer that the degrading of the resolution in the reconstructed images of zebrafish is because high-frequency components can not propagate well in thick samples. To clarify this issue, we added the following descriptions in the section “**Experimental result on imaging biological samples**” of the revised main text: **It is also noticeable that the imaging resolutions of relatively thick biological samples do not look as good as the ones characterized above. Such an observation is likely due to the strong attenuation of high-frequency ultrasound inside thick samples. This fact indicates that for deep-tissue imaging, one may sacrifice the bandwidth of the sensor to gain benefits in other properties.**

7. *What is the demodulation speed of the interrogation means? Can it reach 10 Hz to be consistent with the repetition rate of the pulse laser source in photoacoustic tomography?*

Response: We thank the reviewer for asking this important question, which is critically important for real-time photoacoustic tomography imaging using the proposed interrogation means. As suggested by the reviewer, we supplemented this information in the section “**Discussion**” of the revised main text: **The demodulation speed of the DOFC is another important issue that needs to be considered for real-time imaging. Currently, it took us 0.05 seconds to demodulate an acoustic signal with a length of 10 μ s (large enough to cover the field of view) for the sensor array with 15 elements (Intel(R) Core(TM) i7-10700 CPU @ 2.90GHz and 32 GB RAM). This demodulation speed can catch up with the laser repetition rate of real-time PAT (10 Hz). To incorporate more elements in the future, we may need to optimize the demodulation codes and upgrade the computational facility.**

8. *Linear array may not be the optimal choice for single-shot photoacoustic tomography, as it suffers from the problem of limited view. This problem was alleviated by scanning the samples in the paper but at the cost of multiple shots. I suggest the authors consider the possibility to extend*

this sensor array into a two-dimensional structure. What are the possible challenges to making a two-dimensional sensor array? Can we use the same interrogation means with one coupling waveguide? What is the limiting factor that determines the maximum number of elements?

Response: We thank the reviewer for raising this valuable question. We agree with the reviewer that the capability of extending the current linear array into a two-dimensional structure is of great value to photoacoustic tomography. In the revised submission, we discuss the possibility of extending the linear array into a two-dimensional structure and the potential challenges of using the same interrogation method with one coupling waveguide. In particular, we supplemented this information regarding the practical challenges of making a two-dimensional sensor array and adopting the same interrogation method in the section “**Discussion**” of the revised main text: *Besides the perspective of design, the practical challenge of parallel interrogation also comes from the energy loss of the sensor array. To employ the same interrogation means for the two-dimensional sensor array, a bus waveguide needs to be designed with a relatively long zigzag path to couple all micro-rings distributed in the two-dimensional plane. Given inevitable coupling loss and a 0.2-dB/cm energy loss of the bus waveguide based on the current fabrication technique, these micro-ring sensors will operate in considerably different conditions. The demodulation speed of the DOFC is another important issue that needs to be considered for real-time imaging. Currently, it took us 0.05 seconds to demodulate an acoustic signal with a length of 10 μ s (large enough to cover the field of view) for the sensor array with 15 elements (Intel(R) Core(TM) i7-10700 CPU @ 2.90GHz and 32 GB RAM), which can catch up with the laser repetition rate for real-time photoacoustic tomography (10 Hz). To incorporate more elements, we may need to optimize the demodulation codes and upgrade the computational facility.*

Besides practical considerations, we also added descriptions of the possibilities of making a two-dimensional sensor array and the maximum number of allowed elements from the perspective of design in the section “**Discussion**” of the revised main text, which is detailed in the response to Comment 10.

9. *What is the stability of this sensor array? Since the authors use visible light to tune the resonant frequencies of each microring sensor, I suspect the sensor array may be susceptible to the environment, such as temperature, humidity, and ambient light illumination. The authors should quantify this effect and provide more details on this point.*

Response: We thank the reviewer for asking this important question. The spectrum of the sensor array can be quite stable for several hours, which is sufficiently long to perform PAT. On one hand, we also found that temperature change in water causes resonant frequencies of all micro-ring sensors to shift in the same direction with roughly the same amount, indicating that temperature fluctuations in the water do not scramble the spectrum. On the other hand, we further found that temperature fluctuations in the room do not considerably affect the spectrum of the sensor array. The data that support the above claims, i.e., spectrum stability as a function of time and temperature variation, are supplemented in the revised submission. In particular, in the revised main text, we added the following sentence in the section “**Frequency tuning of the micro-ring sensor array**”: *These*

resonant dips can stay well resolved within 6 hours, which is detailed in Supplementary Note 13.

In the revised Supplementary Information, we also added a new section “Supplementary Note 13” to describe the stability issue in terms of both time and temperature.

Supplementary Note 13. The examination of the spectrum stability of the micro-ring sensor array

The DOFC method strongly relies on the delicate tuning of the sensor spectrum. In this section, we examined the spectrum stability of the micro-ring sensor array. Firstly, we measured the transmission spectrum of the sensor array over time for 6 hours, which is shown in Supplementary Fig. 12. As we can see from the figure, the photo-sensitive effect of the material did cause observable resonant frequency shifts in the spectrum. Nonetheless, all 15 resonant dips are still well-resolvable, allowing sustainable imaging operation using the sensor array. For comparison purposes, grey dashed lines were used to denote the position of the original resonant frequencies. Quantitatively, the mean absolute frequency drifts of the 15 resonant frequencies after 1 hour, 2 hours, 3 hours, 4 hours, 5 hours, and 6 hours are 0.319 GHz, 0.451 GHz, 0.451 GHz, 0.312 GHz, 0.401 GHz, and 0.375 GHz, respectively. Correspondingly, the standard deviation of the frequency drifts of the 15 resonant frequencies after 1 hour, 2 hours, 3 hours, 4 hours, 5 hours, and 6 hours are 0.067 GHz, 0.267 GHz, 0.267 GHz, 0.188 GHz, 0.309 GHz, and 0.208 GHz, respectively. All these values are much smaller than the average separation of adjacent resonant dips (1.66 GHz). These results show that the micro-ring sensor array can still function normally even after being placed in the aqueous environment for 6 hours.

Supplementary Fig. 12. The measured transmission spectra for the micro-ring sensor array over time. The original transmission spectrum (a). Transmission spectra measured after 1 hour (b), 2 hours (c), 3 hours (d), 4 hours (e), 5 hours (f), and 6 hours (g).

In addition, the temperature change of water could also resonant frequency shifts. To examine this issue, we varied the temperature of the aqueous environment within a range from 25 - 40 °C by using a heating pad. Notably, it was observed that all 15 sensors in the array exhibited roughly the same frequency drifts along the same direction. This observation indicates that the variation in the temperature of the water does not scramble the spectrum of the sensor array. The mean resonant frequency shifts for these 15 resonant frequencies are plotted in Supplementary Fig. 13 as a function of temperature using blue dots, while their standard deviations are represented using red circles. As we can see from the figure, the frequency drifts increase linearly with the increased temperature. Through a linear fitting, the measured data results in a thermo-optic coefficient of about 25.9 pm/°C

(blue dashed line), which is in agreement with the one reported in the literature. Moreover, we found that the temperature variation in the room does not considerably change the spectrum of the sensor array.

Supplementary Fig. 13. The measured resonant wavelength shift for micro-ring sensors as a function of temperature. The blue dots and red circles represent the mean values and standard deviations from the measurements of these 15 micro-rings. The blue dashed line denotes the linear fitting of the mean.

10. From the perspective of design, would the authors be able to comment on the scalability of this device? In theory, what would be the maximum number of microrings based on the current configuration? Would it be possible to extend this technique to fabricate a 2D array?

Response: We thank the reviewer for raising this insightful question. In theory, the maximum number of micro-rings based on the current configuration is determined by the detection bandwidth of the DOFC, the quality factor of the micro-ring sensor, and the amount of acoustic pressure induced frequency shifting. We supplemented this information and discussed the possibility of extending this technique to fabricate a two-dimensional array in the section “**Discussion**” of the revised main text: For example, if micro-rings with quality factors up to 2×10^6 can be routinely fabricated, the full width at half maximum of the resonant frequency is about 100 MHz. Considering a ± 100 MHz range for acoustic pressure induced frequency shifting, each micro-ring should at least occupy a frequency bandwidth of 200 MHz in the spectrum. Thus, a 40-GHz detection bandwidth can at most accommodate 200 micro-ring sensors in theory. ..., two-dimensional sensor arrays, including ring shape or bowl shape, are possible to be realized and demonstrated for PAT, ...

Besides theoretical considerations, we also added descriptions of practical challenges in making a two-dimensional sensor array in the revised main text, which is detailed in the response to Comment 8.

In conclusion, the paper presents an impressive step forward, but its value to the community could be enhanced by clearer presenting data.

REVIEWER COMMENTS

Reviewer #1 (Remarks to the Author):

The authors have made major revisions to their manuscript and greatly improved its content. Specifically, the innovation of the current work is much better explained and many of the missing technical details are now provided. All in all, this work is excellent and represents a promising direction towards the goal of fully parallelized chip-based ultrasound detectors using optical technology. Nonetheless, there are still open questions about the specific performance of the device and its underlying mechanism, and I believe that it is essential that the authors address them before the work is accepted.

1) The authors convincingly show that the sensitivity indeed comes from the core and not cladding. However, the results raise a different question: how could this device operate with an air cladding when it is supposed to be submerged (and water absorbs light in the telecom window). My conclusion (which might be wrong) is that in this new characterization experiment, the authors positioned the chip on the water surface, where its top (cladding) was in air and its substrate (SiO_2) was in the water. This means that the impinging acoustic wave came from the direction of the substrate and not the cladding. Is that true for all the measurements? The authors need to add a simple diagram showing the path travelled by the acoustic pulse until it reaches the micro-ring and explain how an air cladding was possible. The reason this is important is that normally the acoustic waves impinge directly on the micro-ring and don't need to travel through the substrate, making the authors' approach unconventional in the field and extremely interesting to other researchers. Also, is there an additional substrate on which the fibers and chip are glued, or is the whole construction held only by the adhesive between the fibers and chip?

2) In Supp. Table 1, the authors should distinguish between the performance achieved by a single detector and an array, for example the higher NEP achieved for the array and lower BW. The reduced BW should also be explicitly mentioned in the text and its value explained (why did they end up with this value and not a different one?).

3) Was the same photodetector used for the single-channel sensor and for the array? If so, what is the reason for the lower sensitivity of the array? Is it that less power was used per channel? If so, how much less? Generally, the compromises that were made to enable parallel detection need to be explained.

4) How much averaging was used in the imaging experiments?

5) Supp. Fig. 7, it would help to show also the signal for the zero angle, which will illustrate how much signal loss there is between angle zero and 10 degrees.

6) Fig. 7h, the PSF is a 3D object with both positive and negative values and the way its currently presented is insufficient. The authors should show all 3 MAP (from all different directions) and slices over the principal axes and not clip out the negative values and artifacts because these are a natural part of any PAT system. I believe this is essential for the readers to fully understand the true nature of

the device. Please do not hide the artifacts in the PSF. They are important and actually contribute to the legitimacy of your technique. If you can show similar artifact in simulation (e.g. due to limited view, or reduced BW as a function of angle) that would be even better. This is really a promising device that could inspire others in the field, and it would be very useful to understand its limitations as well.

7) The authors show that the tuning of the resonance is stable for at least 6 hours and also is stable against temperature variations. That is an excellent result and shows that this approach is compatible with long imaging sessions in a lab environment. However, it is still not clear whether one would need to tune the resonators before each experiment, or could just perform the tuning as part of the fabrication process and continue using the device for days/weeks/months. This is a crucial point about the type of applications this technique is currently useful for, and the possible future need for more stable tuning procedures. Also, if the stability is not too long, it opens up a new challenge for material scientists to find solutions to the drift problem.

8) Seeing the current results, there is a very simple explanation for why the achieved resolutions are lower than the theoretical values. For the axial resolution, the formula used assumed that the BW is the same for all angles, but the detector developed in this work losses its BW at higher angles. So, one would expect some sort of average effective BW that determines that axial resolution, in addition to the explanation of acoustic attenuation. Indeed, higher axial resolutions have been achieved in the past, despite acoustic attenuation. For the lateral resolution, the authors used $f=60$ MHz and $NA=0.5$, but within the 30-degree acceptance angle only the smaller angles had such a central frequency. The average central frequency over that angular range was lower, explaining the lower resolution. A short discussion is in order.

9) Conventionally, when one wishes to quantify the resolution (or more precisely, the width of the PSF) one does that on the envelope of the reconstruction and not only on the positive values. The width of the main lobe just represents the central frequency, but it's the width of the envelope that is inversely proportional to the BW. Obviously, for narrowband sensors, where the PSF has a strong ringing effect with many cycles, the axial resolution is always determined by the envelope width. See for example, [10.1002/jbio.201800357](https://doi.org/10.1002/jbio.201800357). This is true also for any textbook on ultrasound. Accordingly, I suggest updating the number calculated for the axial resolution.

Reviewer #2 (Remarks to the Author):

The revision looks good, and thanks for the detailed responses from the authors. If the authors could help address the following comments, I'd be happy to recommend acceptance.

1. I have a few questions regarding to supplementary Note 13, “the examination of the spectrum stability of the micro-ring sensor array”.
 - a. It seems that the 15 sensors are not equally spaced in frequency domain in supplementary Fig 12. Will it influence the generation and demodulation process of DOFC? In equation (S2), it seems to imply that the comb should be evenly spaced?
 - b. If the uniformity of spacing will influence the working of the device could you provide a tolerable range? If not, could you explain why?
 - c. What is the tolerable range of frequency shift in DOFC? In SI line 356, you mentioned the shift after 6 hr is smaller than the average separation of adjacent resonant dips (1.66 GHz) so the system still functions well. However, the shift is already larger than the bandwidth of the resonant dip. In the typical method (ref. 30), the shift should be smaller than the FWHM of the resonant dip.
 - d. Similarly, could you provide a tolerable range of wavelength shift of standard deviation and wavelength shift in Supplementary Fig. 13.
2. From supplementary Fig.6., is a tunable laser necessary for DFCO? What if one just uses a laser with fixed wavelength?
3. In the new main text line 373, you mentioned that parallel interrogation trades the bandwidth of the detection for compactness of imaging system. What does it mean? What bandwidth is traded?
4. Suggestion: this work has 2 different highlights. One is the high Q, high bandwidth and tunable resonant wavelength microring array sensor. The other is the new parallel interrogation system. For the devices, the manuscript has clear description and characterization. However, for the interrogation system, this is relatively new, and more information will be included and make the manuscript more accessible. E.g. It should include the robustness of the system, subjectivity to noise, and hardware requirements.
5. In main text line 203, the authors mentioned 15 elements are not enough to suppress reconstruction artifacts in PAT and therefore is required to scan. How about the scan for the zebrafish sample? If not, could authors explain why?

Reviewer #3 (Remarks to the Author):

The authors have done a thorough revision to address all the comments of myself and the other two reviewers. It clearly improves the state of the art in the field of photoacoustic imaging.

The experiments, including the newly added ones, are well executed and presented. They support the claims both qualitatively and quantitatively. They are interesting to the fields of both biomedicine and optical engineering.

I support the publication of this work in Nature Communications.

REVIEWER COMMENTS

Reviewer #1 (Remarks to the Author):

The authors have made major revisions to their manuscript and greatly improved its content. Specifically, the innovation of the current work is much better explained and many of the missing technical details are now provided. All in all, this work is excellent and represents a promising direction towards the goal of fully parallelized chip-based ultrasound detectors using optical technology. Nonetheless, there are still open questions about the specific performance of the device and its underlying mechanism, and I believe that it is essential that the authors address them before the work is accepted.

1) The authors convincingly show that the sensitivity indeed comes from the core and not cladding. However, the results raise a different question: how could this device operate with an air cladding when it is supposed to be submerged (and water absorbs light in the telecom window). My conclusion (which might be wrong) is that in this new characterization experiment, the authors positioned the chip on the water surface, where its top (cladding) was in air and its substrate (SiO₂) was in the water. This means that the impinging acoustic wave came from the direction of the substrate and not the cladding. Is that true for all the measurements? The authors need to add a simple diagram showing the path travelled by the acoustic pulse until it reaches the micro-ring and explain how an air cladding was possible. The reason this is important is that normally the acoustic waves impinge directly on the micro-ring and don't need to travel through the substrate, making the authors' approach unconventional in the field and extremely interesting to other researchers. Also, is there an additional substrate on which the fibers and chip are glued, or is the whole construction held only by the adhesive between the fibers and chip?

Response: We thank the reviewer for raising this valuable question. We are sorry that the labeling “air” as the type of cladding confused the reviewer. In the original submission, the device with “air” cladding is a device without any cladding. The operations of this sensor are the same as that of other sensors with claddings, which require the entire sensor to be submerged in water. In this condition, the labeling “air” should actually be “water”. During experiments, acoustic waves directly impinge on the micro-ring and do not need to travel through the substrate in all measurements of this work, which is the same as that in conventional setups. To clarify this point and avoid confusion for potential readers, we added schematics (Supplementary Fig. 4) to illustrate the experimental setup and the following sentences to describe the sensor without cladding in Supplementary Note 5 of the revised Supplementary Information: **A schematic of the experimental setup to examine the effectiveness of the cladding is shown in Supplementary Fig. 4, in which acoustic waves directly impinge on the micro-ring sensors. It should be noted that compared to the transmitted power through micro-ring sensors with claddings, the final output power after transmitting through those micro-ring sensors without cladding decreases by a factor of about 2.3 dB due to water absorption. Thus, for a fair comparison, we adjusted the incident power such that the collected power at the receiver end is at the same level as 0.05 mW.**

Supplementary Fig. 4. Schematics of the experimental setup to examine the effectiveness of the cladding.

Using the same characterization procedure described in the main text to produce Fig. 7(d), ...

Moreover, we changed the label “air” to “water” in the revised Supplementary Fig. 5(c), as follows:

Supplementary Fig. 5. The amplitude maps of the measured signals as a function of time and translational distance. (a) With PDMS cladding. (b) With SiO₂ cladding. (c) Without cladding (water).

Also, the reviewer is correct that there is indeed an additional substrate on which the fibers and chip are glued. To avoid confusion, we modified Supplementary Fig. 6 and revised the corresponding descriptions in Supplementary Note 6 of the revised Supplementary Information: Finally, the entire structure was moved and glued onto a glass slide. This additional glass slide further strengthens the stability of both the fibers and the chip.

Supplementary Fig. 6. A schematic illustration of the fiber-to-chip bonding structure.

2) In Supp. Table 1, the authors should distinguish between the performance achieved by a single detector and an array, for example the higher NEP achieved for the array and lower BW. The reduced BW should also be explicitly mentioned in the text and its value explained (why did they end up with this value and not a different one?).

Response: We thank the reviewer for this valuable suggestion. In the revised Supplementary Tab.1, we added information about the adjusted noise-equivalent pressure (NEP) and bandwidth by using parallel interrogation. Using parallel integration, the NEP spectral density and the NEP are 13.9 mPaHz^{-1/2} and 36.9 Pa, respectively. With parallel integration, the bandwidth is capped by $\Delta f/2$, where Δf is the comb tooth spacing of the DOFC. We supplemented this information to the revised Supplementary Tab. 1 and correspondingly added the following descriptions in Supplementary Note 4 of the revised Supplementary Information for detailed explanation: *It is worth mentioning that the use of a digital optical frequency comb (DOFC) considerably increases the noise level, as discussed in Supplementary Note 11 in detail. Moreover, with parallel interrogation, there is also a compromise in the detecting bandwidth. For DOFC, the comb tooth spacing Δf is inversely proportional to the duration detecting time window T . Since T is the inverse of the sampling rate of the acoustic wave, it can be derived that the sampling rate is the same as the comb tooth spacing Δf . Thus, to meet the requirement of the Nyquist sampling theorem, the largest detecting bandwidth for acoustic waves with parallel interrogation is $\Delta f/2$. In our experiments, $\Delta f = 39.0625$ MHz, leading to a detecting bandwidth of about 20 MHz. Nonetheless, the tunability of the DOFC allows us to conveniently achieve a higher bandwidth by choosing a larger comb tooth spacing, which is at the cost of accuracy in locating resonant frequencies.*

We also modified the description in the section “**Experimental setup for PAT using the sensor array**” of the revised main text: *Moreover, extracting the entire transmission spectrum through the DOFC lowers the sampling rate, which is a compromise to the detecting bandwidth at $\Delta f/2$ (Δf is the comb tooth spacing) and is detailed in Supplementary Note 4.*

We also modified the descriptions of the robustness and noise level of the interrogation system by supplementing more information in the section “**Characterization of the micro-ring sensor**” of the revised main text: *For the same micro-ring sensor, the NEP spectral density and the NEP increase to 13.9 mPaHz^{-1/2} and 36.9 Pa within a 20-MHz bandwidth, respectively (Detailed in Supplementary Note 11).*

3) Was the same photodetector used for the single-channel sensor and for the array? If so, what is the reason for the lower sensitivity of the array? Is it that less power was used per channel? If so, how much less? Generally, the compromises that were made to enable parallel detection need to be explained.

Response: We thank the reviewer for asking this important question. The photodetectors we used for characterizing the performance of a single micro-ring sensor (Agilent, 11982A) and the sensor array with parallel interrogation (Finisar, CPRV1225A) are different. These two detectors have different sensitivities and NEPs. Nonetheless, the difference in hardware is not the major cause for the increased NEP with parallel interrogation. As the reviewer pointed out, the reduced power per channel is the major cause of the increased NEP. In all experiments, regardless of whether employing parallel interrogation or not, the average power being coupled into the sensor array

remains around the level of 0.12 mW to mitigate thermal instability and nonlinear effects. In this condition, each comb tooth can theoretically utilize only $1/N$ of the total power, where N ($=1536$ in this study) is the number of comb teeth used in the DOFC. As a result, the considerably reduced power per each comb tooth is the compromise to enable parallel interrogation. To clarify this point, we added the following sentences in the section “**Characterization of the micro-ring sensor**” of the revised main text: **Compared to the one characterized above, the increased NEPs here are due to the evenly distributed power into each comb tooth ($1/1536$ theoretically), which is a compromise to enable parallel interrogation.**

4) How much averaging was used in the imaging experiments?

Response: We thank the reviewer for asking this important question. In all imaging experiments with biological samples, these samples were rotated with a step size of 1 degree. No additional averaging is required. To clarify this issue, we added the following sentence in the section “**Experimental result on imaging biological samples**” of the revised main text: **No additional averaging was required for for imaging these biological samples.**

5) Supp. Fig. 7, it would help to show also the signal for the zero angle, which will illustrate how much signal loss there is between angle zero and 10 degrees.

Response: We thank the reviewer for this valuable suggestion. In Supplementary Note 9 of the revised Supplementary Information, we modified Supplementary Fig. 8 by including the signals at zero angles. The corresponding descriptions on the labelings were adjusted accordingly.

Supplementary Fig. 8. Characterization for the acceptance angle of the micro-ring sensor. (a)-(e) Time responses of the micro-ring sensor with an acceptance angle at 0° , 10° , 20° , 30° , and 40° , respectively. (f)-(j) Frequency responses of the micro-ring sensor with an acceptance angle at 0° , 10° , 20° , 30° , and 40° , respectively.

6) Fig. 7h, the PSF is a 3D object with both positive and negative values and the way its currently presented in insufficient. The authors should show all 3 MAP (from all different directions) and slices over the principal axes and not clip out the negative values and artifacts because these are a natural part of any PAT system. I believe this is essential for the readers to fully understand the true nature of the device. Please do not hide the artifacts in the PSF. They are important and actually contribute to the legitimacy of your technique. If you can show similar artifact in simulation (e.g.

due to limited view, or reduced BW as a function of angle) that would be even better. This is really a promising device that could inspire others in the field, and it would be very useful to understand its limitations as well.

Response: We thank the reviewer for raising this important issue. In Fig. 7(h) of the original submission, the negative values were directly thresholded out. We agree with the reviewer that these artifacts are a natural part of any PAT system, and thus we modified this figure by keeping these negative values and explicitly showed the artifacts in the revised submission. By keeping negative values, the axial resolution in the revised Fig. 7(j) was re-quantified by using the envelope of the line graph, which exhibited to be $43.6\ \mu\text{m}$. As suggested by the reviewer, we added the following sentences in the section “**Characterization of the micro-ring sensor**” of the revised main text: **One-dimensional profiles for quantifying both the lateral and axial resolutions are illustrated in Figs. 7(i) and (j), and their envelopes exhibiting full width at half maximum of about $50.4\ \mu\text{m}$ and $43.6\ \mu\text{m}$, respectively. These curves have similar shapes to the ones reported in the literature [19, 34, 48, 49] and the ones simulated numerically (detailed in Supplementary Note 12).**

Moreover, as suggested by the reviewer, we used k-wave (Version 1.2.1) to simulate this condition and compare it to the experimentally achieved results. In the revised Supplementary Information, we added a new section “Supplementary Note 12” to describe the simulation results.

Supplementary Note 12 Numerical simulations on the point spread function (PSF) of the imaging system

In this section, we employed a numerical tool (k-wave, Version 1.2.1) to simulate the PSF of the imaging system, which manifests the artifacts due to limited view and reduced bandwidth as a function of angles. The simulation is in a two-dimensional plane and the geometrical parameters adopted in this simulation are identical to the experimental conditions presented in the main text. Furthermore, we used the data presented in Fig. 7(e) of the main text to confine the angular response during simulations. Simulation results along both lateral and axial directions are shown in

Supplementary Figs. 12(a) and (b), exhibiting similar trends as the ones obtained experimentally in Figs. 7(i) and (j), respectively. Moreover, the determined lateral and axial resolutions from experiments and simulations are found to be quantitatively close.

Supplementary Fig. 12. Numerical simulations on the PSF of the imaging system. (a) One-dimensional profile along the lateral direction, suggesting a lateral resolution of 41.2 μm. (b) One-dimensional profile along the axial direction, suggesting an axial resolution of 39.5 μm.

The reviewer also suggested providing a three-dimensional PSF. In this work, two-dimensional PSF was characterized in Fig. 7(g) by imaging the cross-section of a carbon fiber. Since we focus on achieving two-dimensional images with a linear array, we believe two-dimensional PSF can well characterize the imaging system. In general, as the reviewer suggested, three-dimensional PSF could be achieved as well by imaging a point object and scanning the sensor array along the third direction. Nonetheless, we anticipate that the two directions on the transverse plane should exhibit similar behaviors. Due to these reasons, we still keep the present two-dimensional PSF in Fig. 7(h) for convenience.

7) The authors show that the tuning of the resonance is stable for at least 6 hours and also is stable against temperature variations. That is an excellent result and shows that this approach is compatible with long imaging sessions in a lab environment. However, it is still not clear whether one would need to tune the resonators before each experiment, or could just perform the tuning as part of the fabrication process and continue using the device for days/weeks/months. This is a crucial point about the type of applications this technique is currently useful for, and the possible future need for more stable tuning procedures. Also, if the stability is not too long, it opens up a new challenge for material scientists to find solutions to the drift problem.

Response: We thank the reviewer for raising this important question. Although we demonstrated the resonance can be stable for at least 6 hours, it generally cannot be stable for days, weeks, or even months. Thus, for experiments on different days, we still need to tune the sensor array before each experiment. Nonetheless, we anticipate such a problem can be possibly mitigated in the future by adjusting the chemical composition of the chalcogenide glass. To clarify this issue, we added the following sentences in Supplementary Note 14 of the revised Supplementary Information: **However, for experiments on different days, we still need to tune the resonance spectrum of the sensor array before each experiment. For future applications that require long stability, we anticipate that the photosensitive effect can be possibly mitigated by adjusting the chemical composition of the chalcogenide glass, which is out of the scope of this work.**

8) Seeing the current results, there is a very simple explanation for why the achieved resolutions are lower than the theoretical values. For the axial resolution, the formula used assumed that the BW is the same for all angles, but the detector developed in this work losses its BW at higher angles. So, one would expect some sort of average effective BW that determines that axial resolution, in addition to the explanation of acoustic attenuation. Indeed, higher axial resolutions have been achieved in the past, despite acoustic attenuation. For the lateral resolution, the authors used $f=60$ MHz and $NA=0.5$, but within the 30-degree acceptance angle only the smaller angles had such a central frequency. The average central frequency over that angular range was lower, explaining the lower resolution. A short discussion is in order.

Response: We thank the reviewer for helping us explain why the experimentally achieved resolutions are worse than the theoretical values. Following the reviewer's suggestion, we added a short discussion in the section "**Characterization of the micro-ring sensor**" of the revised main text to explain the discrepancy between the experimental values and theoretical predictions:

- (a) The small discrepancy between the estimated value (35.5 μm) and the measured one (50.4 μm) is because $f_0 = 60$ MHz applies for only small acceptance angles so that the average central frequency over the entire angular range is smaller.
- (b) This observation originates from the fact that the bandwidth reduces considerably for large acceptance angles so that the average effective bandwidth over the entire angular range is much smaller. Moreover, the fact that high-frequency components attenuate much more than low-frequency counterparts in agar that covers the carbon fiber might also contribute.

9) Conventionally, when one wishes to quantify the resolution (or more precisely, the width of the PSF) one does that on the envelope of the reconstruction and not only on the positive values. The width of the main lobe just represents the central frequency, but it's the width of the envelope that is inversely proportional to the BW. Obviously, for narrowband sensors, where the PSF has a strong ringing effect with many cycles, the axial resolution is always determined by the envelope width. See for example, [10.1002/jbio.201800357](https://doi.org/10.1002/jbio.201800357). This is true also for any textbook on ultrasound. Accordingly, I suggest updating the number calculated for the axial resolution.

Response: We thank the reviewer for this valuable comment and for providing this useful reference. As suggested by the reviewer, we followed the method described in the provided reference and quantified the axial resolution in Fig. 7(j) by using the envelope, which exhibits to be 43.6 μm . This value was updated in the revised submission.

Reviewer #2 (Remarks to the Author):

The revision looks good, and thanks for the detailed responses from the authors. If the authors could help address the following comments, I'd be happy to recommend acceptance.

1. I have a few questions regarding to supplementary Note 13, “the examination of the spectrum stability of the micro-ring sensor array”.

a. It seems that the 15 sensors are not equally spaced in frequency domain in supplementary Fig 12. Will it influence the generation and demodulation process of DOFC? In equation (S2), it seems to imply that the comb should be evenly spaced?

b. If the uniformity of spacing will influence the working of the device could you provide a tolerable range? If not, could you explain why?

Response: We thank the reviewer for asking these questions regarding the distribution of the resonant frequency and the comb tooth. As the reviewer pointed out, as time evolves, the resonant frequencies of these 15 sensors become non-evenly spaced. Nonetheless, this nonuniformity is not associated with the generation and demodulation process of the DOFC. As shown in Eq. S2, the comb teeth are evenly spaced with very narrow spacing, i.e., much narrower than the spacing between adjacent resonant frequencies. In this condition, each comb tooth effectively acts as an equally spaced sampling point to probe the entire transmission spectrum of the sensor array even when the shifted resonant frequencies become disordered. Therefore, the uniformity of the resonant frequencies of these 15 micro-ring sensors does not influence the normal operation of parallel interrogation with the DOFC. To clarify this issue, we added the following sentences in Supplementary Note 14 of the revised Supplementary Information: **Thanks to the employment of the DOFC, each comb tooth effectively acts as an equally spaced sampling point, serving as a tool to quantify the entire transmission spectrum of the sensor array even when the shifted resonant frequencies become disordered.**

c. What is the tolerable range of frequency shift in DOFC? In SI line 356, you mentioned the shift after 6 hr is smaller than the average separation of adjacent resonant dips (1.66 GHz) so the system still functions well. However, the shift is already larger than the bandwidth of the resonant dip. In the typical method (ref. 30), the shift should be smaller than the FWHM of the resonant dip.

d. Similarly, could you provide a tolerable range of wavelength shift of standard deviation and wavelength shift in Supplementary Fig. 13.

Response: We thank the reviewer for asking these important questions on the tolerable range of wavelength shifts. The means of parallel interrogation used in our work is different from the typical method in Ref. [30]. In the typical method (Ref. [30]), the wavelength or the frequency of the laser is fixed, so that the shift in resonance cannot be larger than the FWHM of the resonant dip. In our work, the employment of DOFC allows us to retrieve the transmission spectrum of the sensor array with a 40 GHz range. Therefore, we only require the resonant frequency of the micro-ring sensor not to shift beyond 40 GHz. Moreover, parallel interrogation of these micro-ring sensor arrays requires adjacent resonant frequencies identifiable. In other words, it means that frequency separation between any two adjacent resonant frequencies should be larger than the FWHM of the resonant dip. No further restriction on the absolute wavelength shift is enforced. To clarify this issue,

we added the following sentences in Supplementary Note 14 of the revised Supplementary Information: **Notably, these wavelength shifts are already larger than the full width at half maximum (FWHM) of typical resonant dips. Such wavelength shifts may cause difficulties for typical conventional methods that generally require the wavelength shift to be smaller than the FWHM of the resonant dip [13]. Nonetheless, the relatively large wavelength shift may not become a big problem with the employment of the DOFC, as it provides a complete transmission spectrum within a 40-GHz range. As long as the resonant frequency does not go beyond this range, we can always precisely locate the position of the resonant frequency. The only criterion applies for parallel interrogation of these micro-ring sensor arrays, which requires adjacent resonant frequencies to be identifiable. In other words, the frequency separation between any two adjacent resonant frequencies should be larger than the FWHM of the resonant dip. No further restriction on the absolute wavelength shifts is enforced.**

2. From supplementary Fig.6., is a tunable laser necessary for DFCO? What if one just uses a laser with fixed wavelength?

Response: We thank the reviewer for asking this important question, which we forgot to emphasize. The reason we draw a tunable laser is that a tunable laser was used for experiments. During experiments, we confirmed that after tuning its operating wavelength to be close to the resonant frequencies of the micro-ring sensors, its wavelength can be fixed thereafter. No additional tunability is required for the generation of DOFC. This operational procedure was discussed in the main text. Thus, as the reviewer pointed out, a laser with a fixed wavelength should work as well.

To avoid confusion to potential readers, since Supplementary Fig. 7 is more like a principle description rather than an experimental description, we modified this figure in the revised Supplementary Information by changing "Tunable laser" to "**Single-frequency laser**".

Correspondingly, we also modified the following sentence in Supplementary Note 7 of the revised Supplementary Information: **As a result, the light output from a single-frequency laser was transformed into a frequency comb.**

3. In the new main text line 373, you mentioned that parallel interrogation trades the bandwidth of the detection for compactness of imaging system. What does it mean? What bandwidth is traded?

Response: We thank the reviewer for asking this important question. The means of parallel interrogation requires measuring a time series to generate the transmission spectrum of the micro-ring sensor array. Thus, the length of the time series restricts the detecting bandwidth of the acoustic waves. For conditions with parallel interrogation, the detecting bandwidth of the acoustic wave is capped by $\Delta f/2$, where Δf is the comb tooth spacing of the DOFC. We supplemented this information to the revised Supplementary Tab. 1 and added the following descriptions in the revised Supplementary Note 4 for detailed explanation correspondingly: **Moreover, with parallel interrogation, there is also a compromise in the detecting bandwidth. For DOFC, the comb tooth spacing Δf is inversely proportional to the duration detecting time window T . Since T is the inverse of the sampling rate of the acoustic wave, it can be derived that the sampling rate is the same as the comb tooth spacing Δf . Thus, to meet the requirement of the Nyquist sampling theorem, the largest detecting bandwidth for acoustic waves with parallel interrogation is $\Delta f/2$. In our experiments, $\Delta f = 39.0625$ MHz, leading to a detecting bandwidth of about 20 MHz. Nonetheless, the tunability of the DOFC allows us to conveniently achieve a higher bandwidth by choosing a larger comb tooth spacing, which is at the cost of accuracy in locating resonant frequencies.**

We also modified the description in the section “**Experimental setup for PAT using the sensor array**” of the revised main text: **Moreover, extracting the entire transmission spectrum through the DOFC lowers the sampling rate, which is a compromise to the detecting bandwidth at $\Delta f/2$ (Δf is the comb tooth spacing) and is detailed in Supplementary Note 4.**

4. Suggestion: this work has 2 different highlights. One is the high Q , high bandwidth and tunable resonant wavelength microring array sensor. The other is the new parallel interrogation system. For the devices, the manuscript has clear description and characterization. However, for the interrogation system, this is relatively new, and more information will be included and make the manuscript more accessible. E.g. It should include the robustness of the system, subjectivity to noise, and hardware requirements.

Response: We thank the reviewer for this valuable suggestion. As suggested by the reviewer, we modified the description of the interrogation system by supplementing more information in the section “**Experimental setup for PAT using the sensor array**” of the revised main text.

- a. **In this work, the interrogation system mainly contains two parts, i.e., the generation and demodulation processes of the DOFC. The generation process started with a digital signal with a multi-carrier bandwidth of 40 GHz and a sequence length of 1,536, which was synthesized through the orthogonal frequency division multiplexing method. The digital signal was then fed into an arbitrary waveform generator (Keysight, M8195A) with a sampling frequency of 60 GHz. In this condition, the corresponding spacing of the subcarrier, i.e., the comb tooth spacing, is 39.0625 MHz (60 GHz/1536). The signal was then amplified and periodically sent to an intensity modulator (Xblue, MXER-LM-20). By setting the intensity modulator with a bias control at the node point, a carrier-suppressed double-sideband modulated DOFC signal was generated with a bandwidth of about 40 GHz (320 pm @1550 nm). Detailed mathematical descriptions of the generation of the DOFC can be found in Supplementary Note 7.**
- b. **To demodulate the DOFC, the measured signal was Fourier transformed into the frequency domain, leading to the transmission spectrum of the sensor array (also detailed in Supplementary Note 7).**

- c. Moreover, extracting the entire transmission spectrum through the DOFC lowers the sampling rate, which is a compromise to the detecting bandwidth at $\Delta f/2$ (Δf is the comb tooth spacing) and is detailed in Supplementary Note 4.

We modified the descriptions of the robustness and noise levels of the interrogation system by supplementing more information in the section “**Characterization of the micro-ring sensor**” of the revised main text.

- a. Compared to the one characterized above, the increased NEPs here are due to the evenly distributed power into each comb tooth (1/1536 theoretically), which is a compromise to enable parallel interrogation.
- b. For the same micro-ring sensor, the NEP spectral density and the NEP increase to 13.9 mPaHz^{1/2} and 36.9 Pa within a 20-MHz bandwidth, respectively (Detailed in Supplementary Note 11).

Besides providing information for those optical and electronic devices, we further added the following sentence to describe the computational facility (hardware) of the interrogation system in the section “**Discussion**” of the revised main text: All signal processing was accomplished on a personal computer with Intel(R) Core (TM) i7-10700 CPU @ 2.90GHz and 32 GB RAM.

5. In main text line 203, the authors mentioned 15 elements are not enough to suppress reconstruction artifacts in PAT and therefore is required to scan. How about the scan for the zebrafish sample? If not, could authors explain why?

Response: We thank the reviewer for asking this important question. When mentioning 15 elements are not enough to suppress the reconstruction artifacts in PAT in the original submission, we aimed to demonstrate the imaging capability of the sensor array in a linear geometry. Without scanning, 15 elements only cover a very small viewing angle, thus being incapable of reconstructing interleaved black hairs in a large area. Therefore, we scanned the sensor array to broaden the viewing angle. As the reviewer pointed out, similar reconstruction artifacts due to limited view in PAT should also occur for imaging zebrafish, if no scanning process was performed. To suppress such artifacts in imaging both leaf vein and zebrafish, instead of scanning the sensor array linearly, we rotated biological samples with a scanning step size of 1 degree. In this condition, the static sensor array can also cover a full view of the samples through the circular imaging geometry. To clarify this issue and avoid confusing potential readers, we modified the following sentences in the section “**Experimental result on imaging biological samples**” of the revised main text: This circular scanning geometry endows the sensor array with a full view of biological samples, which effectively reduces the occurrence of reconstruction artifacts.

Reviewer #3 (Remarks to the Author):

The authors have done a thorough revision to address all the comments of myself and the other two reviewers. It clearly improves the state of the art in the field of photoacoustic imaging.

The experiments, including the newly added ones, are well executed and presented. They support the claims both qualitatively and quantitatively. They are interesting to the fields of both biomedicine and optical engineering.

I support the publication of this work in Nature Communications.

Response: We thank the reviewer for the recognition of our work and truly appreciate for efforts in reviewing this manuscript.

REVIEWER COMMENTS

Reviewer #1 (Remarks to the Author):

The authors have addressed all my comments in their revision. The results are now clear and I believe the reported developments will be of high interest to researchers in the field. I commend the authors for the thorough revision and the excellent quality of the study. I believe this paper will become a classic in the field and recommend its publication.

One last, minor comment: Although some sensitivity was lost going from a single detector to an array, this loss was not major when comparing to other techniques. For example, in a 2022 Optics Letters paper from Y. Hazan et al. (<https://doi.org/10.1364/OL.467652>), parallel readout of arrays using pulse transmission amplitude monitoring involved a two-orders-of-magnitude loss in sensitivity compared to the single-element approach. The authors might want to address this issue to further highlight their achievement.

Reviewer #2 (Remarks to the Author):

Most of the questions are answered. The manuscript can be accepted, and here are a few additional questions and suggestions for authors to consider to improve the paper:

It will be helpful to include a schematic diagram like the one shown in figure 2 (b) of reference [46] to explain the DOFC and the microring resonator.

In line 492, 493, the wavelength and frequency bandwidth doesn't seem to match. Please check.

What is the purpose of the delay line (DL) in the figure 6 (a).

In line 357 in the main text, it mentioned digitizing into 1536 comb teeth. Does it mean that in theory it can support 1536 rings if we don't consider the bandwidth of the ring resonator?

As mentioned in line 361, the comb tooth spacing is 39 MHz. However, the ring frequency width ($Q = 5 \times 10^5$, $\Delta\lambda = 3 \text{ pm}$) is around 370 MHz. Therefore, a ring will contain 10 comb teeth. Will this relatively wide ring resonant width cause any effect? How would you choose the right tooth from the 10 comb teeth within the ring resonance dip?

Point-by-point responses to reviewers' comments

Reviewer #1:

The authors have addressed all my comments in their revision. The results are now clear and I believe the reported developments will be of high interest to researchers in the field. I commend the authors for the thorough revision and the excellent quality of the study. I believe this paper will become a classic in the field and recommend its publication.

Response: We thank the reviewer for the positive comments on our work and truly appreciate his efforts in reviewing and improving this manuscript.

One last, minor comment: Although some sensitivity was lost going from a single detector to an array, this loss was not major when comparing to other techniques. For example, in a 2022 Optics Letters paper from Y. Hazan et al. (<https://doi.org/10.1364/OL.467652>), parallel readout of arrays using pulse transmission amplitude monitoring involved a two-orders-of-magnitude loss in sensitivity compared to the single-element approach. The authors might want to address this issue to further highlight their achievement.

Response: We thank the reviewer for his suggestion. In the section “**Characterization of the micro-ring sensor**” of the revised main text, we highlighted this point of sensitivity loss as suggested by the reviewer: **Moreover, the loss in sensitivity going from a single detector to an array is not large when compared to other techniques. For example, it was shown in Ref. [48] that using pulse transmission amplitude monitoring for parallel readout of arrays could cause orders of magnitude loss in sensitivity compared to the single-element approach.**

Reviewer #2:

Most of the questions are answered. The manuscript can be accepted, and here are a few additional questions and suggestions for authors to consider to improve the paper:

Response: We thank the reviewer for the positive comments on our work. We are also willing to address the remaining questions the reviewer raised to further improve this manuscript.

It will be helpful to include a schematic diagram like the one shown in figure 2 (b) of reference [46] to explain the DOFC and the microring resonator.

Response: We thank the reviewer for making this valuable suggestion. In Fig. 6(b) of the revised main text, we added an inset to schematically describe how to use the DOFC to sample the transmission spectrum of the micro-ring sensor array.

Fig. 6 (a)... (b)... Inset: a schematic diagram that illustrates how the DOFC samples the transmission spectrum of the sensor array. DOFC: digital optical frequency comb.

Corresponding modifications were also made to the revised main text.

In the section “**Experimental setup for PAT using the sensor array.**”: ..., which is measured by the DOFC.

In line 492, 493, the wavelength and frequency bandwidth doesn't seem to match. Please check.

Response: We thank the reviewer for asking for this careful check. In the section “**Frequency tuning of the micro-ring sensor array.**” of the main text, we quantified these values and corrected them as: The average separation between adjacent resonance frequencies is about 1.66 GHz (0.013 nm @ 1550 nm), and these resonant frequencies occupy an overall spectrum range of 24.9 GHz (0.199 nm @ 1550 nm).

What is the purpose of the delay line (DL) in the figure 6 (a).

Response: We thank the reviewer for asking this important question. The purpose of using a delay line is to match the path length difference between the signal beam and the reference beam. During the experiment, we found that without using the delay line, the phase noise of the laser degrades the measurement accuracy. To clarify this point, we added the following sentence in the section

“Experimental setup for PAT using the sensor array.” of the main text: **To avoid the phase noise of the laser and increase measurement accuracy, a delay line was introduced to match the path length difference.**

In line 357 in the main text, it mentioned digitizing into 1536 comb teeth. Does it mean that in theory it can support 1536 rings if we don't consider the bandwidth of the ring resonator?

As mentioned in line 361, the comb tooth spacing is 39 MHz. However, the ring frequency width ($Q = 5 \times 10^5$, $\Delta\lambda = 3 \text{ pm}$) is around 370 MHz. Therefore, a ring will contain 10 comb teeth. Will this relatively wide ring resonant width cause any effect? How would you choose the right tooth from the 10 comb teeth within the ring resonance dip?

Response: We thank the reviewer for raising this important issue, which is worth discussing. In the ideal case, if the resonant frequencies of these micro-ring sensors are always equally spaced and located right at the position of the comb teeth, 1536 comb teeth can theoretically support 1536 micro-ring sensors. However, in practice, the resonant frequencies constantly shift so one needs to use more comb teeth to sample one resonant frequency. Thus, as the reviewer pointed out that we used about ten comb teeth to sample one resonant frequency on average. To locate the position of the resonant frequency, instead of picking one of the ten comb teeth as a representative, we used the values of all ten comb teeth and fit them to a Lorentzian-shaped curve. The position of the fitting curve with the minimum value then became the position of the resonant frequency. In this way, we can locate the position of the resonant frequency accurately, without causing side effects that the reviewer may concern about. To clarify this point, we added the following sentences in the section **“Discussion”** of the revised main text: **In the ideal case, one comb tooth can be used to sample one resonant frequency theoretically. However, during experiments, more comb teeth, i.e., more sampling points, are needed to account for one resonant frequency. Such a choice is because the constantly shifting resonant frequency was not always located right at the position of the comb tooth during experiments. To accurately determine the position of the resonant frequency, one had to collect the values of all the sampling points and fit them to a Lorentzian-shaped curve. The position of the fitting curve with the minimum value then became the position of the resonant frequency.**

REVIEWERS' COMMENTS

Reviewer #2 (Remarks to the Author):

All the questions were answered and I recommend the acceptance of manuscript for publication.

Point-by-point responses to reviewers' comments

Reviewer #2:

All the questions were answered and I recommend the acceptance of manuscript for publication.

Response: We thank the reviewer for recommending our work and truly appreciate the efforts in helping us improve this manuscript.